# Learning One Representation to Optimize All Rewards

**Ahmed Touati**[*]
Mila, University of Montreal
`ahmed.touati@umontreal.ca`

**Yann Ollivier**
Facebook Artificial Intelligence Research
Paris
`yol@fb.com`

## Abstract

We introduce the *forward-backward* (FB) representation of the dynamics of a reward-free Markov decision process. It provides explicit near-optimal policies for any reward specified a posteriori. During an unsupervised phase, we use reward-free interactions with the environment to learn two representations via off-the-shelf deep learning methods and temporal difference (TD) learning. In the test phase, a reward representation is estimated either from reward observations or an explicit reward description (e.g., a target state). The optimal policy for that reward is directly obtained from these representations, with no planning. We assume access to an exploration scheme or replay buffer for the first phase.

The corresponding unsupervised loss is well-principled: if training is perfect, the policies obtained are provably optimal for any reward function. With imperfect training, the sub-optimality is proportional to the unsupervised approximation error. The FB representation learns long-range relationships between states and actions, via a predictive occupancy map, without having to synthesize states as in model-based approaches.

This is a step towards learning controllable agents in arbitrary black-box stochastic environments. This approach compares well to goal-oriented RL algorithms on discrete and continuous mazes, pixel-based MsPacman, and the FetchReach virtual robot arm. We also illustrate how the agent can immediately adapt to new tasks beyond goal-oriented RL. [2]

## 1 Introduction

We consider one kind of unsupervised reinforcement learning problem: Given a Markov decision process (MDP) but no reward information, is it possible to learn and store a compact object that, for any reward function specified later, provides the optimal policy for that reward, with a minimal amount of additional computation? In a sense, such an object would encode in a compact form the solutions of all possible planning problems in the environment. This is a step towards building agents that are fully controllable after first exploring their environment in an unsupervised way.

Goal-oriented RL methods [ACR+17, PAR+18] compute policies for a series of rewards specified in advance (such as reaching a set of target states), but cannot adapt in real time to new rewards, such as weighted combinations of target states or dense rewards.

Learning a model of the world is another possibility, but it still requires explicit planning for each new reward; moreover, synthesizing accurate trajectories of states over long time ranges has proven difficult [Tal17, KST+18].

---

[*]Work done during an internship at Facebook Artificial Intelligence Research Paris.

[2]Code: `https://github.com/ahmed-touati/controllable_agent`

35th Conference on Neural Information Processing Systems (NeurIPS 2021).

Instead, we exhibit an object that is both simpler to learn than a model of the world, and contains the information to recover near-optimal policies for any reward provided a posteriori, without a planning phase.

[BBQ+18] learn optimal policies for all rewards that are linear combinations of a finite number of feature functions provided in advance by the user. This limits applications: e.g., goal-oriented tasks would require one feature per goal state, thus using infinitely many features in continuous spaces. We reuse a policy parameterization from [BBQ+18], but introduce a novel representation with better properties, based on state occupancy prediction instead of expected featurizations. We use theoretical advances on successor state learning from [BTO21]. We obtain the following.

- We prove the existence of a learnable "summary" of a reward-free discrete or continuous MDP, that provides an explicit formula for optimal policies for any reward specified later. This takes the form of a pair of representations $F\colon S \times A \times Z \to Z$ and $B\colon S \times A \to Z$ from state-actions into a representation space $Z \simeq \mathbb{R}^d$, with policies $\pi_z(s) := \arg\max_a F(s, a, z)^\top z$. Once a reward is specified, a value of $z$ is computed from reward values and $B$; then $\pi_z$ is used. Rewards may be specified either explicitly as a function, or as target states, or by samples as in usual RL setups.
- We provide a well-principled unsupervised loss for $F$ and $B$. If FB training is perfect, then the policies are provably optimal for all rewards (Theorem 2). With imperfect training, sub-optimality is proportional to the FB training error (Theorems 8–9). In finite spaces, perfect training is possible with large enough dimension $d$ (Proposition 6).
  Explicitly, $F$ and $B$ are trained so that $F(s, a, z)^\top B(s', a')$ approximates the long-term probability to reach $s'$ from $s$ if following $\pi_z$. This is akin to a model of the environment, without synthesizing state trajectories.
- We provide a TD-like algorithm to train $F$ and $B$ for this unsupervised loss, with function approximation, adapted from recent methods for successor states [BTO21]. No sparse rewards are used: every transition reaches some state $s'$, so every step is exploited. As usual with TD, learning seeks a fixed point but the loss itself is not observable.
- We prove viability of the method on several environments from mazes to pixel-based MsPacman and a virtual robotic arm. For single-state rewards (learning to reach arbitrary states), we provide quantitative comparisons with goal-oriented methods such as HER. (Our method is not a substitute for HER: in principle they could be combined, with HER improving replay buffer management for our method.) For more general rewards, which cannot be tackled a posteriori by trained goal-oriented models, we provide qualitative examples.
- We also illustrate qualitatively the sub-optimalities (long-range behavior is preserved but local blurring of rewards occurs) and the representations learned.

## 2   Problem and Notation

Let $\mathcal{M} = (S, A, P, \gamma)$ be a reward-free Markov decision process with state space $S$ (discrete or continuous), action space $A$ (discrete for simplicity, but this is not essential), transition probabilities $P(s'|s, a)$ from state $s$ to $s'$ given action $a$, and discount factor $0 < \gamma < 1$ [SB18]. If $S$ is finite, $P(s'|s, a)$ can be viewed as a matrix; in general, for each $(s, a) \in S \times A$, $P(\mathrm{d}s'|s, a)$ is a probability measure on $s' \in S$. The notation $P(\mathrm{d}s'|s, a)$ covers all cases.

Given $(s_0, a_0) \in S \times A$ and a policy $\pi\colon S \to \mathrm{Prob}(A)$, we denote $\mathrm{Pr}(\cdot|s_0, a_0, \pi)$ and $\mathbb{E}[\cdot|s_0, a_0, \pi]$ the probabilities and expectations under state-action sequences $(s_t, a_t)_{t \geq 0}$ starting with $(s_0, a_0)$ and following policy $\pi$ in the environment, defined by sampling $s_t \sim P(\mathrm{d}s_t|s_{t-1}, a_{t-1})$ and $a_t \sim \pi(s_t)$.

For any policy $\pi$ and state-action $(s_0, a_0)$, define the *successor measure* $M^\pi(s_0, a_0, \cdot)$ as the measure over $S \times A$ representing the expected discounted time spent in each set $X \subset S \times A$:

$$M^\pi(s_0, a_0, X) := \sum_{t \geq 0} \gamma^t \mathrm{Pr}\left((s_t, a_t) \in X \mid s_0,\, a_0,\, \pi\right) \tag{1}$$

for each $X \subset S \times A$. Viewing $M$ as a measure deals with both discrete and continuous spaces.

Given a reward function $r\colon S \times A \to \mathbb{R}$, the $Q$-function of $\pi$ for $r$ is $Q_r^\pi(s_0, a_0) := \sum_{t \geq 0} \gamma^t \mathbb{E}[r(s_t, a_t)|s_0, a_0, \pi]$. We assume that rewards are bounded, so that all $Q$-functions are well-defined. We state the results for deterministic reward functions, but this is not essential. We abuse notation and write greedy policies as $\pi(s) = \arg\max_a Q(s, a)$ instead of $\pi(s) \in \arg\max_a Q(s, a)$. Ties may be broken any way.

We consider the following informal problem: Given a reward-free MDP $(S, A, P, \gamma)$, can we compute a convenient learnable object $E$ such that, once a reward function $r\colon S \times A \to \mathbb{R}$ is specified, we can easily (with no planning) compute, from $E$ and $r$, a policy $\pi$ whose performance is close to maximal?

## 3 Encoding All Optimal Policies via the Forward-Backward Representation

We first present forward-backward (FB) representations of a reward-free MDP as a way to summarize all optimal policies via explicit formulas. The resulting learning procedure is described in Section 4.

**Core idea.** The main algebraic idea is as follows. Assume, at first, that $S$ is finite. For a fixed policy, the $Q$-function depends linearly on the reward: namely, $Q_r^\pi(s, a) = \sum_{s', a'} M^\pi(s, a, s', a') r(s', a')$ where $M^\pi(s, a, s', a') = \sum_{t \geq 0} \gamma^t \Pr\left((s_t, a_t) = (s', a') | s, a, \pi\right)$. This rewrites as $Q_r^\pi = M^\pi r$ viewing everything as vectors and matrices indexed by state-actions.

Now let $(\pi_z)_{z \in \mathbb{R}^d}$ be any family of policies parameterized by $z$. Assume that for each $z$, we can find $d \times (S \times A)$-matrices $F_z$ and $B$ such that $M^{\pi_z} = F_z^\top B$. Then $Q_r^{\pi_z} = F_z^\top B r$. Specializing to $z_R := Br$, the $Q$-function of policy $\pi_{z_R}$ on reward $r$ is $Q_r^{\pi_{z_R}} = F_{z_R}^\top z_R$. So far $\pi_z$ was unspecified; but if we define $\pi_z(s) := \arg\max_a (F_z^\top z)_{sa}$ at each state $s$, then by definition, $\pi_{z_R}$ is the greedy policy with respect to $F_{z_R}^\top z_R$. At the same time, $F_{z_R}^\top z_R$ is the $Q$-function of $\pi_{z_R}$ for reward $r$: thus, $\pi_{z_R}$ is the greedy policy of its own $Q$-function, and is therefore optimal for reward $r$.

Thus, if we manage to find $F$, $B$, and $\pi_z$ such that $\pi_z = \arg\max F_z^\top z$ and $F_z^\top B = M^{\pi_z}$ for all $z \in \mathbb{R}^d$, then we obtain the optimal policy for any reward $r$, just by computing $Br$ and applying policy $\pi_{Br}$.

This criterion on $(F, B, \pi_z)$ is entirely unsupervised. Since $F$ and $B$ depend on $\pi_z$ but $\pi_z$ is defined via $F$, this is a fixed point equation. An exact solution exists for $d$ large enough (Appendix, Prop. 6), while a smaller $d$ provides lower-rank approximations $M^{\pi_z} \approx F_z^\top B$. In Section 4 we present a well-grounded algorithm to learn such $F$, $B$, and $\pi_z$.

In short, we learn two representations $F$ and $B$ such that $F(s_0, a_0, z)^\top B(s', a')$ is approximately the long-term probability $M^{\pi_z}(s_0, a_0, s', a')$ to reach $(s', a')$ if starting at $(s_0, a_0)$ and following policy $\pi_z$. Then all optimal policies can be computed from $F$ and $B$. We think of $F$ as a representation of the future of a state, and $B$ as the ways to reach a state (Appendix B.4): if $F^\top B$ is large, then the second state is reachable from the first. This is akin to a model of the environment, without synthesizing state trajectories.

**General statement.** In continuous spaces with function approximation, $F_z$ and $B$ become functions $S \times A \to \mathbb{R}^d$ instead of matrices; since $F_z$ depends on $z$, $F$ itself is a function $S \times A \times \mathbb{R}^d \to \mathbb{R}^d$. The sums over states will be replaced with expectations under the data distribution $\rho$.

**Definition 1** (Forward-backward representation). *Let $Z = \mathbb{R}^d$ be a representation space, and let $\rho$ be a measure on $S \times A$. A pair of functions $F\colon S \times A \times Z \to Z$ and $B\colon S \times A \to Z$, together with a parametric family of policies $(\pi_z)_{z \in Z}$, is called a* forward-backward representation *of the MDP with respect to $\rho$, if the following conditions hold for any $z \in Z$ and $(s, a), (s_0, a_0) \in S \times A$:*

$$\pi_z(s) = \arg\max_a F(s, a, z)^\top z, \qquad M^{\pi_z}(s_0, a_0, \mathrm{d}s, \mathrm{d}a) = F(s_0, a_0, z)^\top B(s, a) \rho(\mathrm{d}s, \mathrm{d}a) \quad (2)$$

*where $M^\pi$ is the successor measure defined in (1), and the last equality is between measures.*

**Theorem 2** (FB representations encode all optimal policies). *Let $(F, B, (\pi_z))$ be a forward-backward representation of a reward-free MDP with respect to some measure $\rho$.*

*Then, for any bounded reward function $r\colon S \times A \to \mathbb{R}$, the following holds. Set*

$$z_R := \int_{s, a} r(s, a) B(s, a) \, \rho(\mathrm{d}s, \mathrm{d}a). \quad (3)$$

*assuming the integral exists. Then $\pi_{z_R}$ is an optimal policy for reward $r$ in the MDP. Moreover, the optimal $Q$-function $Q^\star$ for reward $r$ is $Q^\star(s, a) = F(s, a, z_R)^\top z_R$.*

For instance, for a single reward located at state-action $(s, a)$, the optimal policy is $\pi_{z_R}$ with $z_R = B(s, a)$. (In that case the factor $\rho(\mathrm{d}s, \mathrm{d}a)$ does not matter because scaling the reward does not change the optimal policy.)

We present in Section 4 an algorithm to learn FB representations. The measure $\rho$ will be the distribution of state-actions visited in a training set or under an exploration policy: then $z_R = \mathbb{E}_{(s,a)\sim\rho}[r(s,a)B(s,a)]$ can be obtained by sampling from visited states.

In finite spaces, exact FB representations exist, provided the dimension $d$ is larger than $\#S \times \#A$ (Appendix, Prop. 6). In infinite spaces, arbitrarily good approximations can be obtained by increasing $d$, corresponding to a rank-$d$ approximation of the cumulated transition probabilities $M^\pi$. Importantly, the optimality guarantee extends to approximate $F$ and $B$, with optimality gap proportional to $F^\top B - M^{\pi_z}/\rho$ (Appendix, Theorems 8–9 with various norms on $F^\top B - M^\pi/\rho$). For instance, if, for some reward $r$, the error $\left| F(s_0, a_0, z_R)^\top B(s,a) - M^{\pi_{z_R}}(s_0, a_0, \mathrm{d}s, \mathrm{d}a)/\rho(\mathrm{d}s, \mathrm{d}a) \right|$ is at most $\varepsilon$ on average over $(s,a) \sim \rho$ for every $(s_0, a_0)$, then $\pi_{z_R}$ is $3\varepsilon \left\| r \right\|_\infty / (1-\gamma)$-optimal for $r$.

These results justify using some norm over $\left| F^\top B - M^{\pi_z}/\rho \right|$, averaged over $z \in \mathbb{R}^d$, as a training loss for unsupervised reinforcement learning. (Below, we average over $z \in \mathbb{R}^d$ from a fixed rescaled Gaussian. If prior information is available on the rewards $r$, the corresponding distribution of $z_R$ may be used instead.)

If $B$ is fixed in advance and only $F$ is learned, the method has similar properties to successor features based on $B$ (Appendix B.4). But one may set a large $d$ and let $B$ be learned: arguably, by Theorem 2, the resulting features "linearize" optimal policies as much as possible. The features learned in $F$ and $B$ may have broader interest.

## 4 Learning and Using Forward-Backward Representations

Our algorithm starts with an *unsupervised learning phase*, where we learn the representations $F$ and $B$ in a reward-free way, by observing state transitions in the environment, generated from any exploration scheme. Then, in a *reward estimation phase*, we estimate a policy parameter $z_R = \mathbb{E}[r(s,a)B(s,a)]$ from some reward observations, or directly set $z_R$ if the reward is known (e.g., set $z_R = B(s,a)$ to reach a known target $(s,a)$). In the *exploitation phase*, we directly use the policy $\pi_{z_R}(s) = \arg\max_a F(s,a,z_R)^\top z_R$.

**The unsupervised learning phase.** No rewards are used in this phase, and no family of tasks has to be specified manually. $F$ and $B$ are trained off-policy from observed transitions in the environment. The first condition of FB representations, $\pi_z(s) = \arg\max_a F(s,a,z)^\top z$, is just taken as the definition of $\pi_z$ given $F$. In turn, $F$ and $B$ are trained so that the second condition (2), $F(\cdot, z)^\top B = M^{\pi_z}/\rho$, holds for every $z$. Here $\rho$ is the (unknown) distribution of state-actions in the training data. Training is based on the Bellman equation for the successor measure $M^\pi$,

$$M^\pi(s_0, a_0, \{(s', a')\}) = \mathbb{1}_{s_0=s', a_0=a'} + \gamma \mathbb{E}_{s_1 \sim P(\mathrm{d}s_1|s_0, a_0)} M^\pi(s_1, \pi(s_1), \{(s', a')\}). \quad (4)$$

We leverage a well-principled algorithm from [BTO21] in the single-policy setting: it learns the successor measure of a policy $\pi$ without using the sparse reward $\mathbb{1}_{s_0=s', a_0=a'}$ (which would vanish in continuous spaces). Other successor measure algorithms could be used, such as C-learning [ESL21].

The algorithm from [BTO21] uses a parametric model $m_\theta^\pi(s_0, a_0, s', a')$ to represent $M^\pi(s_0, a_0, \mathrm{d}s', \mathrm{d}a') \approx m_\theta^\pi(s_0, a_0, s', a')\rho(\mathrm{d}s', \mathrm{d}a')$. It is not necessary to know $\rho$, only to sample states from it. Given an observed transition $(s_0, a_0, s_1)$ from the training set, generate an action $a_1 \sim \pi(a_1|s_1)$, and sample another state-action $(s', a')$ from the training set, independently from $(s_0, a_0, s_1)$. Then update the parameter $\theta$ by $\theta \leftarrow \theta + \eta\,\delta\theta$ with learning rate $\eta$ and

$$\delta\theta := \partial_\theta m_\theta^\pi(s_0, a_0, s_0, a_0) + \partial_\theta m_\theta^\pi(s_0, a_0, s', a') \times (\gamma\, m_\theta^\pi(s_1, a_1, s', a') - m_\theta^\pi(s_0, a_0, s', a')) \quad (5)$$

This computes the density $m^\pi$ of $M^\pi$ with respect to the distribution $\rho$ of state-actions in the training set. Namely, the true successor state density $m^\pi = M^\pi/\rho$ is a fixed point of (5) in expectation [BTO21, Theorem 6] (and is the only fixed point in the tabular or overparameterized case). Variants exist, such as using a target network for $m_\theta^\pi(s_1, a_1, s', a')$ on the right-hand side, as in DQN.

Thus, we first choose a parametric model $F_\theta, B_\theta$ for the representations $F$ and $B$, and set $m_\theta^{\pi_z}(s_0, a_0, s', a') := F_\theta(s_0, a_0, z)^\top B_\theta(s', a')$. Then we iterate the update (5) over many state-actions and values of $z$. This results in Algorithm 1. At each step, a value of $z$ is picked at random, together with a batch of transitions $(s_0, a_0, s_1)$ and a batch of state-actions $(s', a')$ from the training set, with $(s', a')$ independent from $z$ and $(s_0, a_0, s_1)$.

For sampling $z$, we use a fixed distribution (rescaled Gaussians, see Appendix D). Any number of values of $z$ may be sampled: this does not use up training samples. We use a target network with soft updates (Polyak averaging) as in DDPG. For training we also replace the greedy policy $\pi_z = \arg\max_a F(s, a, z)^\top z$ with a regularized version $\pi_z = \text{softmax}(F(s, a, z)^\top z/\tau)$ with fixed temperature $\tau$ (Appendix D). Since there is unidentifiability between $F$ and $B$ (Appendix, Remark 7), we normalize $B$ via an auxiliary loss in Algorithm 1.

For exploration in this phase, we use the policies being learned: the exploration policy chooses a random value of $z$ from some distribution (e.g., Gaussian), and follows $\pi_z$ for some time (Appendix, Algorithm 1). However, the algorithm can also work from an existing dataset of off-policy transitions.

**The reward estimation phase.** Once rewards are available, we estimate a reward representation (policy parameter) $z_R$ by weighing the representation $B$ by the reward:

$$z_R := \mathbb{E}[r(s, a)B(s, a)] \tag{6}$$

where the expectation must be computed over the same distribution $\rho$ of state-actions $(s, a)$ used to learn $F$ and $B$ (see Appendix B.5 for using a different distribution). Thus, if the reward is black-box as in standard RL algorithms, then the exploration policy has to be run again for some time, and $z_R$ is obtained by averaging $r(s, a)B(s, a)$ over the states visited. An approximate value for $z_R$ still provides an approximately optimal policy (Appendix, Prop. 10 and Thm. 12).

If the reward is known explicitly, this phase is unnecessary. For instance, if the reward is to reach a target state-action $(s_0, a_0)$ while avoiding some forbidden state-actions $(s_1, a_1), ..., (s_k, a_k)$, one may directly set

$$z_R = B(s_0, a_0) - \lambda \sum B(s_i, a_i) \tag{7}$$

where the constant $\lambda$ sets the negative reward for forbidden states and adjusts for the unknown $\rho(\mathrm{d}s_i, \mathrm{d}a_i)$ factors in (3). This can be used for goal-oriented RL.

If the reward is known algebraically as a function $r(s, a)$, then $z_R$ may be computed by averaging the function $r(s, a)B(s, a)$ over a replay buffer from the unsupervised training phase. We may also use a reward model $\hat{r}(s, a)$ of $r(s, a)$ trained on some reward observations from any source.

**The exploitation phase.** Once the reward representation $z_R$ has been estimated, the $Q$-function is estimated as

$$Q(s, a) = F(s, a, z_R)^\top z_R. \tag{8}$$

The corresponding policy $\pi_{z_R}(s) = \arg\max_a Q(s, a)$ is used for exploitation.

Fine-tuning was not needed in our experiments, but it is possible to fine-tune the $Q$-function using actual rewards, by setting $Q(s, a) = F(s, a, z_R)^\top z_R + q_\theta(s, a)$ where the fine-tuning model $q_\theta$ is initialized to 0 and learned via any standard $Q$-learning method.

**Incorporating prior information on rewards in $B$.** Trying to plan in advance for all possible rewards in an arbitrary environment may be too generic and problem-agnostic, and become difficult in large environments, requiring long exploration and a large $d$ to accommodate all rewards. In practice, we are often interested in rewards depending, not on the full state, but only on a part or some features of the state (e.g., a few components of the state, such as the position of an agent, or its neighborhood, rather than the full environment).

If this is known in advance, the representation $B$ can be trained on that part of the state only, with the same theoretical guarantees (Appendix, Theorem 4). $F$ still needs to use the full state as input. This way, the FB model of the transition probabilities (1) only has to learn the future probabilities of the part of interest in the future states, based on the full initial state $(s_0, a_0)$. Explicitly, if $\varphi \colon S \times A \to G$ is a feature map to some features $g = \varphi(s, a)$, and if we know that the reward will be a function $R(g)$, then Theorem 2 still holds with $B(g)$ everywhere instead of $B(s, a)$, and with the successor measure $M^\pi(s_0, a_0, \mathrm{d}g)$ instead of $M^\pi(s_0, a_0, \mathrm{d}s', \mathrm{d}a')$ (Appendix, Theorem 4). Learning is done by replacing $\partial_\theta m_\theta^\pi(s_0, a_0, s_0, a_0)$ with $\partial_\theta m_\theta^\pi(s_0, a_0, \varphi(s_0, a_0))$ in the first term in (5) [BTO21]. Rewards can be arbitrary functions of $g$, so this is more general than [BBQ+18] which only considers rewards linear in $g$. For instance, in MsPacman below, we let $g$ be the 2D position $(x, y)$ of the agent, so we can optimize any reward function that depends on this position.

**Limitations.**   First, this method does not solve exploration: it assumes access to a good exploration strategy. (Here we used the policies $\pi_z$ with random values of $z$, corresponding to random rewards.)

Next, this task-agnostic approach is relevant if the reward is not known in advance, but may not bring the best performance on a particular reward. Mitigation strategies include: increasing $d$; using prior information on rewards by including relevant variables into $B$, as discussed above; and fine-tuning the $Q$-function at test time based on the initial $F^\top B$ estimate.

As reward functions are represented by a $d$-dimensional vector $z_R = \mathbb{E}[r.B]$, some information about the reward is necessarily lost. Any reward uncorrelated to $B$ is treated as 0. The dimension $d$ controls how many types of rewards can be optimized well. A priori, a large $d$ may be required. Still, in the experiments, $d \approx 100$ manages navigation in a pixel-based environment with a huge state space. Appendix B.2 argues theoretically that $d = 2n$ is enough for navigation on an $n$-dimensional grid. The algorithm is linear in $d$, so $d$ can be taken as large as the neural network models can handle.

We expect this method to have an implicit bias for long-range behavior (spatially smooth rewards), while local details of the reward function may be blurred. Indeed, $F^\top B$ is optimized to approximate the successor measure $M^\pi = \sum_t \gamma^t P_\pi^t$ with $P_\pi^t$ the $t$-step transition kernel for each policy $\pi$. The rank-$d$ approximation will favor large eigenvectors of $P_\pi$, i.e., small eigenvectors of the Markov chain Laplacian $\mathrm{Id} -\gamma P_\pi$. These loosely correspond to long-range (low-frequency) behavior [MM07]: presumably, $F$ and $B$ will learn spatially smooth rewards first. Indeed, experimentally, a small $d$ leads to spatial blurring of rewards and $Q$-functions (Fig. 3). Arguably, without any prior information this is a reasonable prior. [SBG17] have argued for the cognitive relevance of low-dimensional approximations of successor representations.

Variance is a potential issue in larger environments, although this did not arise in our experiments. Learning $M^\pi$ requires sampling a state-action $(s_0, a_0)$ and an independent state-action $(s', a')$. In large spaces, most state-action pairs will be unrelated. A possible mitigation is to combine FB with strategies such as Hindsight Experience Replay [ACR+17] to select goals related to the current state-action. The following may help a lot: the update of $F$ and $B$ decouples as an expectation over $(s_0, a_0)$, times an expectation over $(s', a')$. Thus, by estimating these expectations by a moving average over a dataset, it is easy to have many pairs $(s_0, a_0)$ interact with many $(s', a')$. The cost is handling full $d \times d$ matrices. This will be explored in future work.

## 5   Experiments

We first consider the task of reaching arbitrary goal states. For this, we can make quantitative comparisons to existing goal-oriented baselines. Next, we illustrate qualitatively some tasks that cannot be tackled a posteriori by goal-oriented methods, such as introducing forbidden states. Finally, we illustrate some of the representations learned.

### 5.1   Environments and Experimental Setup

We run our experiments on a selection of environments that are diverse in term of state space dimensionality, stochasticity and dynamics.

- Discrete Maze is the classical gridworld with four rooms. States are represented by one-hot unit vectors.
- Continuous Maze is a two dimensional environment with impassable walls. States are represented by their Cartesian coordinates $(x, y) \in [0, 1]^2$. The execution of one of the actions moves the agent in the desired direction, but with normal random noise added to the position of the agent.
- FetchReach is a variant of the simulated robotic arm environment from [PAR+18] using discrete actions instead of continuous actions. States are 10-dimensional vectors consisting of positions and velocities of robot joints.
- Ms. Pacman is a variant of the Atari 2600 game Ms. Pacman, where an episode ends when the agent is captured by a monster [RUMS18]. States are obtained by processing the raw visual input directly from the screen. Frames are preprocessed by cropping, conversion to grayscale and downsampling to $84 \times 84$ pixels. A state $s_t$ is the concatenation of $(x_{t-12}, x_{t-8}, x_{t-4}, x_t)$ frames, i.e. an $84 \times 84 \times 4$ tensor. An action repeat of 12 is used. As Ms. Pacman is not originally a multi-goal domain, we define the goals as the 148 reachable coordinates $(x, y)$ on the screen; these can be reached only by learning to avoid monsters.

For all environments, we run algorithms for 800 epochs, with three different random seeds. Each epoch consists of 25 cycles where we interleave between gathering some amount of transitions, to add to the replay buffer, and performing 40 steps of stochastic gradient descent on the model parameters. To collect transitions, we generate episodes using some behavior policy. For both mazes, we use a uniform policy while for FetchReach and Ms. Pacman, we use an $\varepsilon$-greedy policy with respect to the current approximation $F(s, a, z)^\top z$ for a sampled $z$. At evaluation time, $\varepsilon$-greedy policies are also used, with a smaller $\varepsilon$. More details are given in Appendix D.

## 5.2 Goal-Oriented Setting: Quantitative Comparisons

We investigate the FB representation over goal-reaching tasks and compare it to goal-oriented baselines: DQN[3], and DQN with HER when needed. We define sparse reward functions. For Discrete Maze, the reward function is equal to one when the agent's state is equal exactly to the goal state. For Discrete Maze, we measured the quality of the obtained policy to be the ratio between the true expected discounted reward of the policy for its goal and the true optimal value function, on average over all states. For the other environments, the reward function is equal to one when the distance of the agent's position and the goal position is below some threshold, and zero otherwise. We assess policies by computing the average success rate, i.e the average number of times the agent successfully reaches its goal.

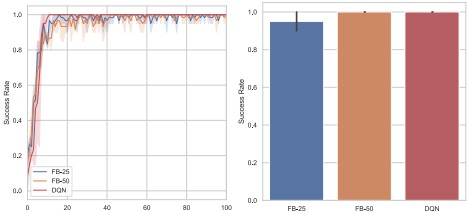

Figure 1: Comparative performance of FB for different dimensions and DQN in FetchReach. **Left**: success rate averaged over 20 randomly selected goals as function of the first 100 training epochs. **Right**: success rate averaged over 20 random goals after 800 training epochs.

Figure 2: Comparative performance of FB for different dimensions and DQN in Ms. Pacman. **Left**: success rate averaged over 20 randomly selected goals as function of the first 200 training epochs. **Right**: success rate averaged over the goal space after 800 training epochs.

Figs. 1 and 2 show the comparative performance of FB for different dimensions $d$, and DQN respectively in FetchReach and Ms. Pacman (similar results in Discrete and Continuous Mazes are provided in Appendix D). In Ms. Pacman, DQN totally fails to learn and we had to add HER to make it work. The performance of FB consistently increases with the dimension $d$ and the best dimension matches the performance of the goal-oriented baseline.

In Discrete Maze, we observe a drop of performance for $d = 25$ (Appendix D, Fig. 8): this is due to the spatial smoothing induced by the small rank approximation and the reward being nonzero only if the agent is exactly at the goal. This spatial blurring is clear on heatmaps for $d = 25$ vs $d = 75$ (Fig. 3). With $d = 25$ the agent often stops right next to its goal.

To evaluate the sample efficiency of FB, after each epoch, we evaluate the agent on 20 randomly selected goals. Learning curves are reported in Figs. 1 and 2 (left). In all environments, we observe no loss in sample efficiency compared to the goal-oriented baseline. In Ms. Pacman, FB even learns faster than DQN+HER.

## 5.3 More Complex Rewards: Qualitative Results

We now investigate FB's ability to generalize to new tasks that cannot be solved by an already trained goal-oriented model: reaching a goal with forbidden states imposed a posteriori, reaching the nearest of two goals, and choosing between a small, close reward and a large, distant one.

First, for the task of reaching a target position $g_0$ ☆ while avoiding some forbidden positions $g_1, \dots g_k$ ●, we set $z_R = B(g_1) - \lambda \sum_{i=1}^{k} B(g_i)$ and run the corresponding $\varepsilon$-greedy policy defined

---

[3]Here DQN is short for goal-oriented DQN, $Q(s, a, g)$.

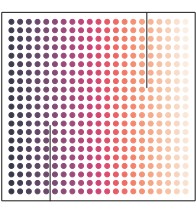 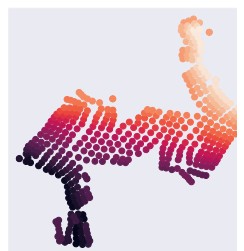 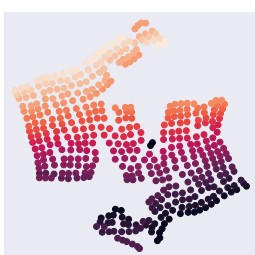

Figure 7: Visualization of FB embedding vectors on Continuous Maze after projecting them in two-dimensional space with t-SNE. **Left**: the states to be mapped. **Middle**: the $F$ embedding. **Right**: the $B$ embedding. The walls appear as large dents; the smaller dents correspond to the number of steps needed to get past a wall.

by $F(s,a,z_R)^\top z_R$. Fig. 5 shows the resulting trajectories, which succeed at solving the task for the different domains. In Ms. Pacman, the path is suboptimal (though successful) due to the sudden appearance of a monster along the optimal path. (We only plot the initial frame; see the full series of frames along the trajectory in Appendix D, Fig. 16.) Fig. 4 (left) provides a contour plot of $\max_{a\in A} F(s,a,z_R)^\top z_R$ for the continuous maze and shows the landscape shape around the forbidden regions.

Next, we consider the task of reaching the closest target among two equally rewarding positions $g_0$ and $g_1$, by setting $z_R = B(g_0) + B(g_1)$. The optimal $Q$-function is *not* a linear combination of the $Q$-functions for $g_0$ and $g_1$. Fig. 6 shows successful trajectories generated by the policy $\pi_{z_R}$. On the contour plot of $\max_{a\in A} F(s,a,z_R)^\top z_R$ in Fig. 4 (right), the two rewarding positions appear as basins of attraction. Similar results for a third task are shown in Appendix D: introducing a "distracting" small reward next to the initial position of the agent, with a larger reward further away.

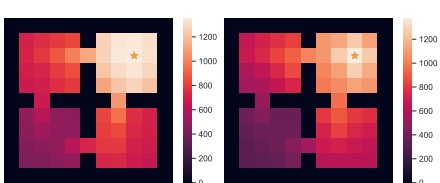

Figure 3: Heatmap of $\max_a F(s,a,z_R)^\top z_R$ for $z_R = B(\star)$ **Left**: $d=25$. **Right**: $d=75$.

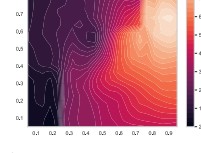 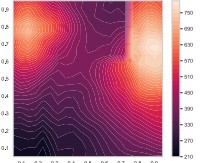

Figure 4: Contour plot of $\max_{a\in A} F(s,a,z_R)^\top z_R$ in Continuous Maze. **Left**: for the task of reaching a target while avoiding a forbidden region, **Right**: for two equally rewarding targets.

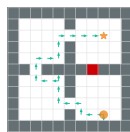 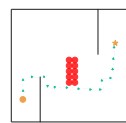 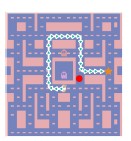

Figure 5: Trajectories generated by the $F^\top B$ policies for the task of reaching a target position (star shape $\star$ while avoiding forbidden positions (red shape ●)

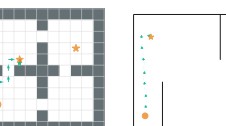 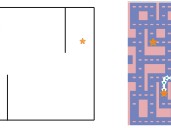

Figure 6: Trajectories generated by the $F^\top B$ policies for the task of reaching the closest among two equally rewarding positions (star shapes $\star$). (Optimal $Q$-values are not linear over such mixtures.)

## 5.4 Embedding Visualizations

We visualize the learned FB state embeddings for Continuous Maze by projecting them into 2-dimensional space using t-SNE [VdMH08] in Fig. 7. For the forward embeddings, we set $z=0$ corresponding to the uniform policy. We can see that FB partitions states according to the topology induced by the dynamics: states on opposite sides of walls are separated in the representation space

and states on the same side lie together. Appendix D includes embedding visualizations for different $z$ and for Discrete Maze and Ms. Pacman.

# 6 Related work

[BBQ$^+$18] learn optimal policies for rewards that are linear combinations of a finite number of feature functions provided in advance by the user. This approach cannot tackle generic rewards or goal-oriented RL: this would require introducing one feature per possible goal state, requiring infinitely many features in continuous spaces.

Our approach does not require user-provided features describing the future tasks, thanks to using successor *states* [BTO21] where [BBQ$^+$18] use successor *features*. Schematically, and omitting actions, successor features start with user-provided features $\varphi$, then learn $\psi$ such that $\psi(s_0) = \sum_{t \geq 0} \gamma^t \mathbb{E}[\varphi(s_t) \mid s_0]$. This limits applicability to rewards that are linear combinations of $\varphi$. Here we use successor *state* probabilities, namely, we learn two representations $F$ and $B$ such that $F(s_0)^\top B(s') = \sum_{t \geq 0} \gamma^t \Pr(s_t = s' \mid s_0)$. This does not require any user-provided input.

Thus we learn two representations instead of one. The learned backward representation $B$ is absent from [BBQ$^+$18]. $B$ plays a different role than the user-provided features $\varphi$ of [BBQ$^+$18]: if the reward is known a priori to depend only on some features $\varphi$, we learn $B$ on top of $\varphi$, which represents all rewards that depend linearly or nonlineary on $\varphi$. Up to a change of variables, [BBQ$^+$18] is recovered by setting $B = \mathrm{Id}$ on top of $\varphi$, or $B = \varphi$ and $\varphi = \mathrm{Id}$, and then only training $F$.

We use a similar parameterization of policies by $F(s, a, z)^\top z$ as in [BBQ$^+$18], for similar reasons, although $z$ encodes a different object.

Successor representations where first defined in [Day93] for finite spaces, corresponding to an older object from Markov chains, the fundamental matrix [KS60, Bré99, GS97]. [SBG17] argue for their relevance for cognitive science. For successor representations in continuous spaces, a finite number of features $\varphi$ are specified first; this can be used for generalization within a family of tasks, e.g., [BDM$^+$17, ZSBB17, GHB$^+$19, HDB$^+$19]. [BTO21] moves from successor features to successor states by providing pointwise occupancy map estimates even in continuous spaces, without using the sparse reward $\mathbb{1}_{s_t = s'}$. We borrow a successor state learning algorithm from [BTO21]. [BTO21] also introduced simpler versions of $F$ and $B$ for a single, fixed policy; [BTO21] does not consider the every-optimal-policy setting.

There is a long literature on goal-oriented RL. For instance, [SHGS15] learn goal-dependent value functions, regularized via an explicit matrix factorization. Goal-dependent value functions have been investigated in earlier works such as [FD02] and [SMD$^+$11]. Hindsight experience replay (HER) [ACR$^+$17] improves the sample efficiency of multiple goal learning with sparse rewards. A family of rewards has to be specified beforehand, such as reaching arbitrary target states. Specifying rewards a posteriori is not possible: for instance, learning to reach target states does not extend to reaching the nearest among several goals, reaching a goal while avoiding forbidden states, or maximizing any dense reward.

Hierarchical methods such as options [SPS99] can be used for multi-task RL problems. However, policy learning on top of the options is still needed after the task is known.

For finite state spaces, [JKSY20] use reward-free interactions to build a training set that summarizes a finite environment, in the sense that any optimal policies later computed on this training set instead of the true environment are provably $\varepsilon$-optimal, for any reward. They prove tight bounds on the necessary set size. Policy learning still has to be done afterwards for each reward.

**Acknowledgments.** The authors would like to thank Léonard Blier, Diana Borsa, Alessandro Lazaric, Rémi Munos, Tom Schaul, Corentin Tallec, Nicolas Usunier, and the anonymous reviewers for numerous comments, technical questions, references, and invaluable suggestions for presentation that led to an improved text.

# 7 Conclusion

The forward-backward representation is a learnable mathematical object that "summarizes" a reward-free MDP. It provides near-optimal policies for any reward specified a posteriori, without planning. It is learned from black-box reward-free interactions with the environment. In practice, this unsupervised method performs comparably to goal-oriented methods for reaching arbitrary goals, but is also able to tackle more complex rewards in real time. The representations learned encode the MDP dynamics and may have broader interest.

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
