## Outline of the Supplementary Material

The Appendix is organized as follows.

- Appendix A presents the pseudo-code of the unsupervised phase of FB algorithm.
- Appendix B provides extended theoretical results on approximate solutions and general goals:
  - Section B.1 formalizes the forward-backward representation with a goal or feature space.
  - Section B.2 establishes the existence of exact FB representations in finite spaces, and discusses the influence of the dimension $d$.
  - Section B.3 shows how approximate solutions provide approximately optimal policies.
  - Section B.4 shows how $F$ and $B$ are successor and predecessor features of each other, and how the policies are optimal for rewards linearly spanned by $B$.
  - Section B.5 explains how to estimate $z_R$ at test time from a state distribution different from the training distribution.
  - Section B.6 presents a note of the measure $M^\pi$ and its density $m^\pi$.
- Appendix C provides proofs of all theoretical results above.
- Appendix D provides additional information about our experiments:
  - Section D.1 describes the environments.
  - Section D.2 describes the different architectures used for FB as well as the goal-oriented DQN.
  - Section D.3 provides implementation details and hyperparameters.
  - Section D.4 provides additional experimental results.

# A Algorithm

The unsupervised phase of the FB algorithm is described in Algorithm 1.

The loss function for the Bellman equation on $F$ and $B$ appears on line 19, while the auxiliary loss function for orthonormalization of $B$ appears on line 21.

---

**Algorithm 1** FB algorithm: Unsupervised Phase

---

1: **Inputs:** replay buffer $\mathcal{D}$, Polyak coefficient $\alpha$, $\nu$ a probability distribution over $\mathbb{R}^d$, randomly initialized networks $F_\theta$ and $B_\omega$, learning rate $\eta$, mini-batch size $b$, number of episodes $E$, number of gradient updates $N$, temperature $\tau$ and regularization coefficient $\lambda$.

2: **for** $m = 1, \ldots$ **do**

3:    /* Collect $E$ episodes

4:    **for** episode $e = 1, \ldots E$ **do**

5:      Sample $z \sim \nu$

6:      Observe an initial state $s_0$

7:      **for** $t = 1, \ldots$ **do**

8:        Select an action $a_t$ according to some behaviour policy (e.g the $\varepsilon$-greedy with respect to $F_\theta(s_t, a, z)^\top z$ )

9:        Observe next state $s_{t+1}$

10:        Store transition $(s_t, a_t, s_{t+1})$ in the replay buffer $\mathcal{D}$

11:      **end for**

12:    **end for**

13:    /* Perform $N$ stochastic gradient descent updates

14:    **for** $n = 1 \ldots N$ **do**

15:      Sample a mini-batch of transitions $\{(s_i, a_i, s_{i+1})\}_{i \in I} \subset \mathcal{D}$ of size $|I| = b$.

16:      Sample a mini-batch of target state-action pairs $\{(s_i', a_i')\}_{i \in I} \subset \mathcal{D}$ of size $|I| = b$.

17:      Sample a mini-batch of $\{z_i\}_{i \in I} \sim \nu$ of size $|I| = b$.

18:      Set $\pi_{z_i}(\cdot \mid s_{i+1}) = \texttt{softmax}(F_{\theta^-}(s_{i+1}, \cdot, z_i)^\top z_i / \tau)$

19:      $\mathcal{L}(\theta, \omega) =$
$$\frac{1}{2b^2} \sum_{i,j \in I^2} \left( F_\theta(s_i, a_i, z_i)^\top B_\omega(s_j', a_j') - \gamma \sum_{a \in A} \pi_{z_i}(a \mid s_{i+1}) \cdot F_{\theta^-}(s_{i+1}, a, z_i)^\top B_{\omega^-}(s_j', a_j') \right)^2 - \frac{1}{b} \sum_{i \in I} F_\theta(s_i, a_i, z_i)^\top B_\omega(s_i, a_i)$$

20:      /* Compute orthonormality regularization loss

21:      $\mathcal{L}_{\texttt{reg}}(\omega) = \frac{1}{b^2} \sum_{i,j \in I^2} B_\omega(s_i, a_i)^\top \texttt{stop-gradient}(B_\omega(s_j', a_j')) \cdot \texttt{stop-gradient}(B_\omega(s_i, a_i)^\top B_\omega(s_j', a_j')) - \frac{1}{b} \sum_{i \in I} B_\omega(s_i, a_i)^\top \texttt{stop-gradient}(B_\omega(s_i, a_i))$

22:      Update $\theta \leftarrow \theta - \eta \nabla_\theta \mathcal{L}(\theta, \omega)$ and $\omega \leftarrow \omega - \eta \nabla_\omega(\mathcal{L}(\theta, \omega) + \lambda \cdot \mathcal{L}_{\texttt{reg}}(\omega))$

23:    **end for**

24:    /* Update target network parameters

25:    $\theta^- \leftarrow \alpha \theta^- + (1 - \alpha)\theta$

26:    $\omega^- \leftarrow \alpha \omega^- + (1 - \alpha)\omega$

27: **end for**

---

# B    Extended Results: Approximate Solutions and General Goals

**Notation.** *In general, we denote by $M^\pi$ the successor measure of policy $\pi$ as defined in (1), and by $m^\pi$ its density, if it exists, with respect to a reference measure $\rho$. Namely,*

$$M^\pi(s_0, a_0, \mathrm{d}s, \mathrm{d}a) = m^\pi(s_0, a_0, s, a)\rho(\mathrm{d}s, \mathrm{d}a). \tag{9}$$

*Thus, the defining property of forward-backward representations (Definition 1) is $F(s_0, a_0, z)^\top B(s, a) = m^{\pi_z}(s_0, a_0, s, a)$.*

*We use the same convention for parametric models, with $M_\theta^\pi$ a measure and $m_\theta^\pi$ its density. The reference measure $\rho$ is fixed and may be unknown (typically $\rho$ is the distribution of state-actions in a training set or under an exploration policy).*

## B.1    The Forward-Backward Representation With a Goal or Feature Space

Here we state a generalization of Theorem 2 covering some extensions mentioned in the text.

First, this covers rewards known to only depend on certain features $g = \varphi(s, a)$ of the state-action $(s, a)$, where $\varphi$ is a known function with values in some goal state $G$ (for instance, rewards depending only on some components of the state). Then it is enough to compute $B$ as a function of the goal $g$. Theorem 2 corresponds to $\varphi = \mathrm{Id}$. This is useful to introduce prior information when available, resulting in a smaller model $(F, B)$.

This also recovers successor features as in [BBQ+18], defined by user-provided features $\varphi$. Indeed, fixing $B$ to Id and setting our $\varphi$ to the $\varphi$ of [BBQ+18] (or fixing $B$ to their $\varphi$ and our $\varphi$ to Id) will represent the same set of rewards and policies as in [BBQ+18], namely, optimal policies for rewards linear in $\varphi$ (although with a slightly different learning algorithm and up to a linear change of variables for $F$ and $z$ given by the covariance of $\varphi$, see Appendix B.4). More generally, keeping the same $\varphi$ but letting $B$ free (with larger $d$) can provide optimal policies for rewards that are arbitrary functions of $\varphi$, linear or not.

For this, we extend successor state measures to values in goal spaces, representing the discounted time spent at each goal by the policy. Namely, given a policy $\pi$, let $M^\pi$ be the the successor state measure of $\pi$ over goals $g$:

$$M^\pi(s, a, \mathrm{d}g) := \sum_{t \geq 0} \gamma^t \Pr\left(\varphi(s_t, a_t) \in \mathrm{d}g \mid s_0 = s,\, a_0 = a,\, \pi\right) \tag{10}$$

for each state-action $(s, a)$ and each measurable set $\mathrm{d}g \subset G$. This will be the object approximated by $F(s, a, z)^\top B(g)$.

Second, we use a more general model of successor states: instead of $m \approx F^\top B$ we use $m \approx F^\top B + \bar{m}$ where $\bar{m}$ does not depend on the action, so that the $F^\top B$ part only computes advantages. This lifts the constraint that the model of $m$ has rank at most $d$, because there is no restriction on the rank on $\bar{m}$: the rank restriction only applies to the advantage function.

Third, we state a form of policy improvement for the FB representation. Namely, the $Q$-function $F(s, a, z_R)^\top z_R$ for a given reward can be computed as a supremum over all values of $z$.

For simplicity we state the result with deterministic rewards, but this extends to stochastic rewards, because the expectation $z_R$ will be the same.

**Definition 3** (Extended forward-backward representation of an MDP). *Consider an MDP with state space $S$ and action space $A$. Let $\varphi\colon S \times A \to G$ be a function from state-actions to some goal space $G = \mathbb{R}^k$.*

*Let $Z = \mathbb{R}^d$ be some representation space. Let*

$$F\colon S \times A \times Z \to Z, \qquad B\colon G \to Z, \qquad \bar{m}\colon S \times Z \times G \to \mathbb{R} \tag{11}$$

*be three functions. For each $z \in Z$, define the policy*

$$\pi_z(a|s) := \arg\max_a F(s, a, z)^\top z. \tag{12}$$

*Let $\rho$ be any measure over the goal space $G$.*

*We say that $F$, $B$, and $\bar{m}$ are an* extended forward-backward representation *of the MDP with respect to $\rho$, if the following holds: for any $z \in Z$, any state-actions $(s,a)$, and any goal $g \in G$, one has*

$$M^{\pi_z}(s, a, \mathrm{d}g) = \left(F(s, a, z)^\top B(g) + \bar{m}(s, z, g)\right) \rho(\mathrm{d}g) \tag{13}$$

*where $M^{\pi_z}$ is the successor state measure (10) of policy $\pi_z$.*

**Theorem 4** (Forward-backward representation of an MDP, with features as goals)**.** *Consider an MDP with state space $S$ and action space $A$. Let $\varphi\colon S \times A \to G$ be a function from state-actions to some goal space $G = \mathbb{R}^k$.*

*Let $F$, $B$, and $\bar{m}$ be an extended forward-backward representation of the MDP with respect to some measure $\rho$ over $G$.*

*Then the following holds. Let $R\colon S \times A \to \mathbb{R}$ be any bounded reward function, and assume that this reward function depends only on $g = \varphi(s, a)$, namely, that there exists a function $r\colon G \to \mathbb{R}$ such that $R(s, a) = r(\varphi(s, a))$. Set*

$$z_R := \int_{g \in G} r(g)B(g)\,\rho(\mathrm{d}g) \tag{14}$$

*assuming the integral exists.*

*Then:*

1. *$\pi_{z_R}$ is an optimal policy for reward $R$ in the MDP.*

2. *For any $z \in Z$, the $Q$-function of policy $\pi_z$ for the reward $R$ is equal to*

$$Q^{\pi_z}(s, a) = F(s, a, z)^\top z_R + \bar{V}^z(s) \tag{15}$$

   *and the optimal $Q$-function $Q_R^\star$ is obtained when $z = z_R$:*

$$Q_R^\star(s, a) = F(s, a, z_R)^\top z_R + \bar{V}^{z_R}(s). \tag{16}$$

   *Here*

$$\bar{V}^z(s) := \int_{g \in G} \bar{m}(s, z, g)r(g)\,\rho(\mathrm{d}g) \tag{17}$$

   *and in particular $\bar{V} = 0$ if $\bar{m} = 0$.*

   *The advantages $Q^{\pi_z}(s, a) - Q^{\pi_z}(s, a')$ do not depend on $\bar{V}$, so computing $\bar{V}$ is not necessary to obtain the policies.*

3. *If $\bar{m} = 0$, then for any state-action $(s, a)$ one has*

$$Q_R^\star(s, a) = F(s, a, z_R)^\top z_R = \sup_{z \in Z} F(s, a, z)^\top z_R. \tag{18}$$

(We do not claim that $\bar{V}$ is the value function and $F^\top z_R$ the advantage function, only that the sum is the $Q$-function. When $\bar{m} = 0$, the term $F^\top z_R$ is the whole $Q$-function.)

The last point of the theorem is a form of policy improvement. Indeed, by the second point, $F(s, a, z)^\top z_R$ is the estimated $Q$-function of policy $\pi_z$ for rewards $r$. This may be useful if $z_R$ falls outside of the training distribution for $F$: then the values of $F(s, a, z_R)$ may not be safe to use. In that case, it may be useful to use a finite set $Z' \subset Z$ of values of $z$ closer to the training distribution, and use the estimate $\sup_{z \in Z'} F(s, a, z)^\top z_R$ instead of $F(s, a, z_R)^\top z_R$ for the optimal $Q$-function. A similar option has been used, e.g., in [BBQ+18], but in the end it was not necessary in our experiments.

**Remark 5.** *Formally, the statement holds for arbitrary $\rho$, but it only makes sense if $\rho$ has full support (or at least covers all reachable parts of the state space): (13) requires the support of $M^{\pi_z}$ to be included in that of $\rho$. Otherwise, FB representations may not exist and the statement is empty.*

### B.2 Existence of Exact $FB$ Solutions, Influence of Dimension $d$, Uniqueness

**Existence of exact $FB$ representations in finite spaces.** We now prove existence of an exact solution for finite spaces if the representation dimension $d$ is at least $\#S \times \#A$. Solutions are never unique: one may always multiply $F$ by an invertible matrix $C$ and multiply $B$ by $(C^\top)^{-1}$, see Remark 7 below (this allows us to impose orthonormality of $B$ in the experiments).

The constraint $d \geq \#S \times \#A$ can be largely overestimated depending on the tasks of interest, though. For instance, we prove below that in an $n$-dimensional toric grid $S = \{1, \ldots, k\}^n$, $d = 2n$ is enough to obtain optimal policies for reaching every target state (a set of tasks smaller than optimizing all possible rewards).

**Proposition 6** (Existence of an exact $FB$ representation for finite state spaces)**.** *Assume that the state and action spaces $S$ and $A$ of an MDP are finite. Let $Z = \mathbb{R}^d$ with $d \geq \#S \times \#A$. Let $\rho$ be any measure on $S \times A$, with $\rho(s, a) > 0$ for any $(s, a)$.*

*Then there exists $F \colon S \times A \times Z \to Z$ and $B \colon S \times A \to Z$, such that $F^\top B$ is equal to the successor state density of $\pi_z$ with respect to $\rho$:*

$$F^\top(s, a, z)B(s', a') = \sum_{t \geq 0} \gamma^t \frac{\Pr((s_t, a_t) = (s', a') \mid s_0 = s, \, a_0 = a, \, \pi_z)}{\rho(s', a')} \tag{19}$$

*for any $z \in Z$ and any state-actions $(s, a)$ and $(s', a')$, where $\pi_z$ is defined as in Definition 1 by $\pi_z(s) = \arg\max_a F(s, a, z)^\top z$.*

**A small dimension $d$ can be enough for navigation: examples.** In practice, even a small $d$ can be enough to get optimal policies for reaching arbitrary many states (as opposed to optimizing all possible rewards). Let us give an example with $S$ a toric $n$-dimensional grid of size $k$.

Let us start with $n = 1$. Take $S = \{0, \ldots, k - 1\}$ to be a length-$k$ cycle with three actions $a \in \{-1, 0, 1\}$ (go left, stay in place, go right). Take $d = 2$, so that $Z = \mathbb{R}^2 \simeq \mathbb{C}$.

We consider the tasks of reaching an arbitrary target state $s'$, for every $s' \in S$. Thus the goal state is $G = S$ in the notation of Theorem 4, and $B$ only depends on $s'$. The policy for such a reward is $\pi_{z_R} = \pi_{B(s')}$.

For a state $s \in \{0, \ldots, k - 1\}$ and action $a \in \{-1, 0, 1\}$, define

$$F(s, a, z) := e^{2i\pi(s+a)/k}, \qquad B(s) := e^{2i\pi s/k}. \tag{20}$$

Then one checks that $\pi_{B(s')}$ is the optimal policy for reaching $s'$, for every $s' \in S$. Indeed, $F(s, a, z_R)^\top z_R = \cos(2\pi(s + a - s')/k)$. This is maximized for the action $a$ that brings $s$ closer to $s'$.

So the policies will be optimal for reaching every target $s' \in S$, despite the dimension being only 2.

By taking the product of $n$ copies of this example, this also works on the $n$-dimensional toric grid $S = \{0, \ldots, k - 1\}^n$ with $2n + 1$ actions (add $\pm 1$ in each direction or stay in place), with a representation of dimension $d = 2n$ in $\mathbb{C}^n$, namely, by taking $B(s)_j := e^{2i\pi s_j/k}$ for ecah direction $j$ and likewise for $F$. Then $\pi_{B(s')}$ is the optimal policy for reaching $s'$ for every $s' \in S$.

More generally, if one is only interested in the optimal policies for reaching states, then it is easy to show that there exist functions $F \colon S \times A \to Z$ and $B \colon S \to Z$ such that the policies $\pi_z$ describe the optimal policies to reach each state: it is enough that $B$ be injective (typically requiring $d = \dim(S)$). Indeed, for any state $s \in S$, let $\pi_s^\star$ be the optimal policy to reach $s$. We want $\pi_z$ to be equal to $\pi_s^\star$ for $z = B(s)$ (the value of $z_R$ for a reward located at $s$). This translates as $\arg\max_a F(s', a, B(s))^\top B(s) = \pi_s^\star(s')$ for every other state $s'$. This is realized just by letting $F$ be any function such that $F(s', \pi_s^\star(s'), B(s)) := B(s)$ and $F(s', a, B(s)) := -B(s)$ for every other action $a$. As soon as $B$ is injective, there exists such a function $F$. (Unfortunately, we are not able to show that the learning algorithm reaches such a solution.)

Let us turn to uniqueness of $F$ and $B$.

**Remark 7.** *Let $C$ be an invertible $d \times d$ matrix. Given $F$ and $B$ as in Theorem 2, define*

$$B'(s,a) := CB(s,a), \qquad F'(s,a,z) := (C^\top)^{-1}F(s,a,C^{-1}z) \tag{21}$$

*together with the policies $\pi'_z(s) := \arg\max_a F'(s,a,z)^\top z$. For each reward $r$, define $z'_R := \mathbb{E}_{(s,a)\sim\rho}[r(s,a)B'(s,a)]$.*

*Then this operation does not change the policies or estimated Q-values: for any reward, we have $\pi'_{z'_R} = \pi_{z_R}$, and $F'(s,a,z'_R)^\top z'_R = F(s,a,z_R)^\top z_R$.*

*In particular, assume that the components of $B$ are linearly independent. Then, taking $C = \left(\mathbb{E}_{(s,a)\sim\rho} B(s,a)B(s,a)^\top\right)^{-1/2}$, $B'$ is $L^2(\rho)$-orthonormal. So up to reducing the dimension $d$ to $\mathrm{rank}(B)$, we can always assume that $B$ is $L^2(\rho)$-orthonormal, namely, that $\mathbb{E}_{(s,a)\sim\rho} B(s,a)B(s,a)^\top = \mathrm{Id}$.*

Reduction to orthonormal $B$ will be useful in some proofs below. Even after imposing that $B$ be orthonormal, solutions are not unique, as one can still apply a rotation matrix on the variable $z$.

For a single policy $\pi_z$, the $F^\top B$ decomposition may be further standardized in several ways: after a linear change of variables in $\mathbb{R}^d$, and up to decreasing $d$ by removing unused directions in $\mathbb{R}^d$, one may assume either that $\mathrm{Cov}_\rho B = \mathrm{Id}$ and $\mathrm{Cov}_\rho F$ is diagonal, or that $\mathrm{Cov}_\rho F = \mathrm{Cov}_\rho B$ is diagonal, or write the decomposition as $\tilde{F}^\top D \tilde{B}$ with $D$ diagonal and $\mathrm{Cov}_\rho \tilde{F} = \mathrm{Cov}_\rho \tilde{B} = \mathrm{Id}$, thus corresponding to an approximation of the singular value decomposition of the successor measure $M^{\pi_z}$ in $L^2(\rho)$. However, since $B$ is shared between all values of $z$ and all policies $\pi_z$, it is a priori not possible to realize this for all $z$ simultaneously.

## B.3 Approximate Solutions Provide Approximately Optimal Policies

Here we prove that the optimality in Theorems 2 and 4 is robust to approximation errors during training: approximate solutions still provide approximately optimal policies. We deal first with the approximation errors on $(F, B)$ during unsupervised training, then on $z_R$ during the reward estimation phase (in case the reward is not known explicity).

### B.3.1 Influence of Approximate $F$ and $B$: Optimality Gaps are Proportional to $m^\pi - F^\top B$

In continuous spaces, Theorems 2 and 4 are somewhat spurious: the equality $F^\top B = m$ that defines FB representations will never hold exactly with finite representation dimension $d$. Instead, $F^\top B$ will only be a rank-$d$ approximation of $m$. Even in finite spaces, since $F$ and $B$ are learned by a neural network, we can only expect that $F^\top B \approx m$ in general. Therefore, we provide results that extend Theorems 2 and 4 to approximate training and approximate FB representations.

Optimality gaps are directly controlled by the error $m^\pi - F^\top B$ on the solution $F^\top B$. We provide this result for different notions of approximations between $m^\pi$ and $F^\top B$. First, in sup norm over $(s, a)$ but in expectation over $(s', a')$ (so that a perfect model of the successor states $(s', a')$ of each $(s, a)$ is not necessary, only an average model). Second, for the weak topology on measures (this is the most relevant in continuous spaces: for instance, a Dirac measure can be approached by a continuous model in the weak topology). Finally, we provide pointwise estimates instead of only norms: for any reward, we show that the optimality gaps at each state can be bounded by an explicit matrix product directly involving the FB error matrix $m^\pi - F^\top B$ (Theorem 9).

$F$ and $B$ are trained such that $F(s,a,z)^\top B(s',a')$ approximates the successor state density $m^{\pi_z}(s, a, s', a')$. In the simplest case, we prove that if for some reward $R$,

$$\mathbb{E}_{(s',a')\sim\rho}\left|F(s,a,z_R)^\top B(s',a') - m^{\pi_{z_R}}(s,a,s',a')\right| \le \varepsilon \tag{22}$$

for every $(s, a)$, then the optimality gap of policy $\pi_{z_R}$ is at most $(3\varepsilon/(1-\gamma))\sup|R|$ for that reward (Theorem 8, first case).

In continuous spaces, $m^\pi$ is usually a distribution (Appendix B.6), so such an approximation will not hold, and it is better to work on the measures themselves rather than their densities, namely, to compare $F^\top B \rho$ to $M^\pi$ instead of $F^\top B$ to $m^\pi$. We prove that if $F^\top B \rho$ is close to $M^\pi$ in the weak topology, then the resulting policies are optimal for any Lipschitz reward.[4]

---

[4]This also holds for continuous rewards, but the Lipschitz assumption yields an explicit bound in Theorem 8.

Remember that a sequence of nonnegative measures $\mu_n$ converges weakly to $\mu$ if for any bounded, continuous function $f$, $\int f(x)\mu_n(dx)$ converges to $\int f(x)\mu(dx)$ ([Bog07], §8.1). The associated topology can be defined via the following *Kantorovich–Rubinstein* norm on nonnegative measures ([Bog07], §8.3)

$$\|\mu - \mu'\|_{\mathrm{KR}} := \sup\left\{\left|\int f(x)\mu(\mathrm{d}x) - \int f(x)\mu'(\mathrm{d}x)\right| : f \text{ 1-Lipschitz function with } \sup|f| \le 1\right\} \tag{23}$$

where we have equipped the state-action space with any metric compatible with its topology.[5]

The following theorem states that if $F^\top B$ approximates the successor state density of the policy $\pi_z$ (for various sorts of approximations), then $\pi_z$ is approximately optimal. Given a reward function $r$ on state-actions, we denote

$$\|r\|_\infty := \sup_{(s,a)\in S\times A} |r(s,a)| \tag{24}$$

and

$$\|r\|_{\mathrm{Lip}} := \sup_{(s,a)\ne(s',a')\in S\times A} \frac{r(s,a) - r(s',a')}{d((s,a),(s',a'))} \tag{25}$$

where we have chosen any metric on state-actions.

The first statement is for any bounded reward. The second statement only assumes an $F^\top B$ approximation in the weak topology but only applies to Lipschitz rewards. The third statement is more general and is how we prove the first two: weaker assumptions on $F^\top B$ work on a stricter class of rewards.

**Theorem 8** (If $F$ and $B$ approximate successor states, then the policies $\pi_z$ yield approximately optimal returns). *Let $F\colon S \times A \times Z \to Z$ and $B\colon S \times A \to Z$ be any functions, and define the policy $\pi_z(s) = \arg\max_a F(s,a,z)^\top z$ for each $z \in Z$.*

*Let $\rho$ be any positive probability distribution on $S \times A$, and for each policy $\pi$, let $m^\pi$ be the density of the successor state measure $M^\pi$ of $\pi$ with respect to $\rho$. Let*

$$\hat{m}^z(s,a,s',a') := F(s,a,z)^\top B(s',a'), \qquad \hat{M}^z(s,a,\mathrm{d}s',\mathrm{d}a') := \hat{m}^z(s,a,s',a')\rho(\mathrm{d}s',\mathrm{d}a') \tag{26}$$

*be the estimates of $m$ and $M$ obtained via the model $F$ and $B$.*

*Let $r\colon S \times A \to \mathbb{R}$ be any bounded reward function. Let $V^\star$ be the optimal value function for this reward $r$. Let $\hat{V}^{\pi_z}$ be the value function of policy $\pi_z$ for this reward. Let $z_R = \mathbb{E}_{(s,a)\sim\rho}[r(s,a)B(s,a)]$.*

*Then:*

1. *If $\mathbb{E}_{(s',a')\sim\rho}|\hat{m}^{z_R}(s,a,s',a') - m^{\pi_{z_R}}(s,a,s',a')| \le \varepsilon$ for any $(s,a)$ in $S \times A$, then $\left\|\hat{V}^{\pi_{z_R}} - V^\star\right\|_\infty \le 3\varepsilon\|r\|_\infty/(1-\gamma)$.*

2. *If $r$ is Lipschitz and $\left\|\hat{M}^{z_R}(s,a,\cdot) - M^{\pi_{z_R}}(s,a,\cdot)\right\|_{\mathrm{KR}} \le \varepsilon$ for any $(s,a) \in S \times A$, then $\left\|\hat{V}^{\pi_{z_R}} - V^\star\right\|_\infty \le 3\varepsilon\max(\|r\|_\infty, \|r\|_{\mathrm{Lip}})/(1-\gamma)$.*

3. *More generally, let $\|\cdot\|_A$ be a norm on functions and $\|\cdot\|_B$ a norm on measures, such that $\int f\,\mathrm{d}\mu \le \|f\|_A\|\mu\|_B$ for any function $f$ and measure $\mu$. Then for any reward function $r$ such that $\|r\|_A < \infty$,*

$$\left\|\hat{V}^{\pi_{z_R}} - V^\star\right\|_\infty \le \frac{3\|r\|_A}{1-\gamma}\sup_{s,a}\left\|\hat{M}^{z_R}(s,a,\cdot) - M^{\pi_{z_R}}(s,a,\cdot)\right\|_B. \tag{27}$$

*Moreover, the optimal $Q$-function is close to $F(s,a,z_R)^\top z_R$:*

$$\sup_{s,a}\left|F(s,a,z_R)^\top z_R - Q^\star(s,a)\right| \le \frac{2\|r\|_A}{1-\gamma}\sup_{s,a}\left\|\hat{M}^{z_R}(s,a,\cdot) - M^{\pi_{z_R}}(s,a,\cdot)\right\|_B. \tag{28}$$

---

[5]The Kantorovich–Rubinstein norm is closely related to the $L^1$ Wasserstein distance on probability distributions, but slightly more general as it does not require the distance functions to be integrable: the Wasserstein distance metrizes weak convergence among those probability measures such that $\mathbb{E}[d(x,x_0)] < \infty$.

**Pointwise optimality gaps.** We now turn to a more precise estimation of the optimality gap at each state, expressed directly as a matrix product involving the FB error $m^\pi - F^\top B$.

Here we assume that the state space is finite: this is not essential but simplifies notation since everything can be represented as matrices and vectors. For each stochastic policy $\pi\colon S \to \mathrm{Prob}(A)$, we denote $P_\pi$ the associated stochastic matrix on state-actions:

$$(P_\pi)_{(sa)(s'a')} := P(s'|s,a)\pi(s')[a']. \tag{29}$$

We also view rewards and $Q$-functions as vectors indexed by state-actions.

**Theorem 9.** *Assume that the state space is finite. Let $F\colon S \times A \times \mathbb{R}^d \to \mathbb{R}^d$ and $B\colon S \times A \to \mathbb{R}^d$ be any functions. Define $\pi_z$ as in Definition 1. Let $\rho > 0$ be any probability distribution over state-actions, and let $m^\pi$ be the successor density (9) of policy $\pi$ with respect to $\rho$.*

*For each $z \in \mathbb{R}^d$, define the FB error $E(z)$ as a matrix over state-actions:*

$$E(z)_{(sa)(s'a')} := m^{\pi_z}(s,a,s',a') - F(s,a,z)^\top B(s',a'). \tag{30}$$

*Let $r$ be any reward function and let $Q^\star$ and $\pi^\star$ be its optimal $Q$-function and policy. Let $z_R = \mathbb{E}_\rho[r.B]$ as in Theorem 2, and let $Q^{\pi_{z_R}}$ be the $Q$-function of policy $\pi_{z_R}$.*

*Then we have the componentwise inequality between state-action vectors*

$$0 \le Q^\star - Q^{\pi_{z_R}} \le \left(\sum_{t \ge 0} \gamma^{t+1} P_{\pi^\star}^t\right)\left(P_{\pi^\star} - P_{\pi_{z_R}}\right) E(z_R)\,\mathrm{diag}(\rho)\,r. \tag{31}$$

*In particular, if $E(z_R) = 0$ then $\pi_{z_R}$ is optimal for reward $r$.*

The matrix $\sum_{t \ge 0} \gamma^{t+1} P_{\pi^\star}^t$ represents states visited along the optimal trajectory starting at the initial state. Multiplying by $(P_{\pi^\star} - P_{\pi_{z_R}})$ visits states one step away from this trajectory. Therefore, what matters are the values of the FB error matrix $E(z_R)_{(sa)(s'a')}$ at state-actions $(s,a)$ one step away from the optimal trajectories.

In unsupervised RL, we have no control over the first part $\sum_{t \ge 0} \gamma^{t+1} P_{\pi^\star}^t$, which depends only on the optimal trajectories of the unknown future tasks $r$. Therefore, in general it makes sense to just minimize $E(z)$ in some matrix norm, for as many values of $z$ as possible. (The choice of matrix norm influences which rewards will be best optimized, as illustrated by Theorem 8.)

### B.3.2 An approximate $z_R$ yields an approximately optimal policy

We now turn to the second source of approximation: computing $z_R$ in the reward estimation phase. This is a problem only if the reward is not specified explicitly.

We deal in turn with the effect of using a model of the reward function, and the effect of estimating $z_R = \mathbb{E}_{(s,a)\sim\rho}\, r(s,a)B(s,a)$ via sampling.

Which of these options is better depends a lot on the situation. If training of $F$ and $B$ is perfect, by construction the policies are optimal for each $z_R$: thus, estimating $z_R$ from rewards sampled at $N$ states $(s_i, a_i) \sim \rho$ will produce the optimal policy for exactly that empirical reward, namely, a nonzero reward at each $(s_i, a_i)$ but zero everywhere else, thus overfitting the reward function. Reducing the dimension $d$ reduces this effect, since rewards are projected on the span of the features in $B$: $B$ plays both the roles of a transition model and a reward regularizer. This appears as a $\sqrt{d/N}$ factor in Theorem 12 below.

Thus, if both the number of samples to train $F$ and $B$ and the number of reward samples are small, using a smaller $d$ will regularize both the model of the environment and the model of the reward. However, if the number of samples to train $F$ and $B$ is large, yielding an excellent model of the environment, but the number of reward samples is small, then learning a model of the reward function will be a better option than direct empirical estimation of $z_R$.

The first result below states that reward misidentification comes on top of the approximation error of the $F^\top B$ model. This is relevant, for instance, if a reward model $\hat{r}$ is estimated by an external model using some reward values.

**Proposition 10** (Influence of estimating $z_R$ by an approximate reward). *Assume $\rho$ is a probability distribution. Let $\hat{r} \colon S \times A \to \mathbb{R}$ be any reward function. Let $\hat{z}_R = \mathbb{E}_{(s,a)\sim\rho}[\hat{r}(s,a)B(s,a)]$.*

*Let $\varepsilon_{FB}$ be the error attained by the $F^\top B$ model in Theorem 8 for reward $\hat{r}$; namely, assume that $\left\|\hat{V}^{\pi_{\hat{z}_R}} - \hat{V}^\star\right\|_\infty \leq \varepsilon_{FB}$ with $\hat{V}^\star$ the optimal value function for $\hat{r}$.*

*Then the policy $\pi_{\hat{z}_R}(s) = \arg\max_a F(s,a,\hat{z}_R)^\top \hat{z}_R$ defined by the model $\hat{r}$ is $\left( \dfrac{2\,\|r - \hat{r}\|_\infty}{(1-\gamma)} + \varepsilon_{FB} \right)$-optimal for reward $r$.*

This still assumes that the expectation $\hat{z}_R = \mathbb{E}_{(s,a)\sim\rho}[\hat{r}(s,a)B(s,a)]$ is computed exactly for the model $\hat{r}$. If $\hat{r}$ is given by an explicit model, this expectation can in principle be computed on the whole replay buffer used to train $F$ and $B$, so variance would be low. Nevertheless, we provide an additional statement which covers the influence of the variance of the estimator of $z_R$, whether this estimator uses an external model $\hat{r}$ or a direct empirical average reward observations $r(s,a)B(s,a)$.

**Definition 11.** *The skewness $\zeta(B)$ of $B$ is defined as follows. Assume $B$ is bounded. Let $B_1, \ldots, B_d \colon S \times A \to \mathbb{R}$ be the functions of $(s,a)$ defined by each component of $B$. Let $\langle B \rangle$ be the linear span of the $(B_i)_{1 \leq i \leq d}$ as functions on $S \times A$. Set*

$$\zeta(B) := \sup_{f \in \langle B \rangle,\, f \neq 0} \frac{\|f\|_\infty}{\|f\|_{L^2(\rho)}}. \tag{32}$$

**Theorem 12** (Influence of estimating $z_R$ by empirical averages). *Assume that $\rho$ is a probability distribution. Assume that $z_R = \mathbb{E}_{(s,a)\sim\rho}[r(s,a)B(s,a)]$ is estimated via*

$$\hat{z}_R := \frac{1}{N}\sum_{i=1}^{N} \hat{r}_i B(s_i, a_i) \tag{33}$$

*using $N$ independent samples $(s_i, a_i) \sim \rho$, where the $r_i$ are random variables such that $\mathbb{E}[\hat{r}_i|s_i, a_i] = r(s_i, a_i)$, $\mathrm{Var}[\hat{r}_i|s_i, a_i] \leq v$ for some $v \in \mathbb{R}$, and the $\hat{r}_i$ are mutually independent given $(s_i, a_i)_{i=1,\ldots,N}$.*

*Let $V^\star$ be the optimal value function for reward $r$, and let $\hat{V}$ be the value function of the estimated policy $\pi_{\hat{z}_R}$ for reward $r$.*

*Then, for any $\delta > 0$, with probability at least $1 - \delta$,*

$$\left\|\hat{V} - V^\star\right\|_\infty \leq \varepsilon_{FB} + \frac{2}{1-\gamma}\sqrt{\frac{\zeta(B)\,d}{N\delta}\left(v + \|r(s,a) - \mathbb{E}_\rho\, r\|^2_{L^2(\rho)}\right)} \tag{34}$$

*which is therefore the bound on the optimality gap of $\pi_{\hat{z}_R}$ for $r$. Here $\varepsilon_{FB}$ is the error due to the $F^\top B$ model approximation, defined as in Proposition 10.*

The proofs are not direct, because $F$ is not continuous with respect to $z$. Contrary to $Q$-values, successor states are not continuous in the reward: if an action has reward 1 and the reward for another action changes from $1-\varepsilon$ to $1+\varepsilon$, the return values change by at most $2\varepsilon$, but the actions and states visited by the optimal policy change a lot. So it is not possible to reason by continuity on each of the terms involved.

### B.4   $F$ and $B$ as Successor and Predecessor Features of Each Other, Optimality for Rewards in the Span of $B$

We now give two statements. The first encodes the idea that $F$ encodes the future of a state while $B$ encodes the past of a state.

The second proves that the $FB$ policies are optimal for any reward that lies in the linear span of the features learned in $B$; for rewards out of this span, it is best if the features in $B$ are spatially smooth.

The intuition that $F$ and $B$ encode the future and past of states is formalized as follows: if $F$ and $B$ minimize their unsupervised loss, then $F$ is equal to the successor features from the dual features of $B$, and $B$ is equal to the *predecessor* features from the dual features of $F$ (Theorem 13).

This statement holds for a fixed $z$ and the corresponding policy $\pi_z$. So, for the rest of this section, $z$ is fixed. In this section, we also assume that $\rho$ is a probability distribution.

By "dual" features we mean the following. Define the $d \times d$ covariance matrices

$$\text{Cov } F := \mathbb{E}_{(s,a) \sim \rho}[F(s,a,z)F(s,a,z)^\top], \qquad \text{Cov } B := \mathbb{E}_{(s,a) \sim \rho}[B(s,a)B(s,a)^\top]. \qquad (35)$$

Then $(\text{Cov } F)^{-1/2}F(s,a,z)$ is $L^2(\rho)$-orthonormal and likewise for $B$. The "dual" features are $(\text{Cov } F)^{-1}F(s,a,z)$ and $(\text{Cov } B)^{-1}B(s,a)$, without the square root: these are the least square solvers for $F$ and $B$ respectively, and these are the ones that appear below.

The unsupervised forward-backward loss for a fixed $z$ is

$$\ell(F, B) := \int \left| F(s,a,z)^\top B(s',a') - \sum_{t \geq 0} \gamma^t \frac{P_t(\mathrm{d}s', \mathrm{d}a'|s,a,\pi_z)}{\rho(\mathrm{d}s', \mathrm{d}a')} \right|^2 \rho(\mathrm{d}s, \mathrm{d}a)\rho(\mathrm{d}s', \mathrm{d}a') \qquad (36)$$

$$= \left\| F(\cdot, z)^\top B(\cdot) - m^{\pi_z}(\cdot, \cdot) \right\|^2_{L^2(\rho) \otimes L^2(\rho)}. \qquad (37)$$

Thus, minimizers in dimension $d$ correspond to an SVD of the successor state density in $L^2(\rho)$, truncated to the largest $d$ singular values.

**Theorem 13.** *Consider a smooth parametric model for $F$ and $B$, and assume this model is overparameterized.* [6] *Also assume that the data distribution $\rho$ has positive density everywhere.*

*Let $z \in Z$. Assume that for this $z$, $F$ and $B$ lie in $L^2(\rho)$ and achieve a local extremum of $\ell(F, B)$ within this parametric model. Namely, the derivative $\partial_\theta \ell(F, B)$ of the loss with respect to the parameters $\theta$ of $F$ is $0$, and likewise for $B$.*

*Then $F$ is equal to $(\text{Cov } B)^{-1}$ times the successor features of $B$: for any $(s,a) \in S \times A$,*

$$(\text{Cov } B)F(s,a,z) = \sum_{t \geq 0} \gamma^t \int_{(s',a')} P_t(\mathrm{d}s', \mathrm{d}a'|s,a,\pi_z) \, B(s',a') \qquad (38)$$

*and $B$ is equal to $(\text{Cov } F)^{-1}$ times the predecessor features of $F$:*

$$(\text{Cov } F)B(s',a') = \sum_{t \geq 0} \gamma^t \int_{(s,a)} \frac{P_t(\mathrm{d}s', \mathrm{d}a'|s,a,\pi_z)}{\rho(\mathrm{d}s', \mathrm{d}a')} \, F(s,a,z) \, \rho(\mathrm{d}s, \mathrm{d}a) \qquad (39)$$

*$\rho$-almost everywhere. Here the covariances have been defined in (35), and $P_t(\cdot|s,a,\pi)$ denotes the law of $(s_t, a_t)$ under trajectories starting at $(s,a)$ and following policy $\pi$.*

*The same result holds when working with features $\varphi(s', a')$, just by applying it to $B \circ \varphi$.*

Note that in the FB framework, we may normalize either $F$ or $B$ (Remark 7), but not both.

As a consequence of Theorem 13, we characterize below which kind of rewards we can capture if we fix $B$ and train only $F$ to minimize the unsupervised loss given $B$. Namely, we show that for any reward $r$, the resulting policy is optimal for the $L^2(\rho)$-orthogonal projection of $r$ onto the span of $B$.

**Theorem 14** (Optimizing $F$ for a given $B$; influence of the span of $B$). *Let $B$ a fixed function in $L^2(\rho, \mathbb{R}^d)$. Define the* span *of $B$ as the set of functions $w^\top B \in L^2(\rho, \mathbb{R})$ when $w$ ranges in $\mathbb{R}^d$.*

*Consider a smooth parametric model for $F$, and assume this model is overparameterized. Assume that $F$ lies in $L^2(\rho)$ and achieves a local extremum of $\ell(F, B)$ within this parametric model for each $z \in Z$.*

*Assume that the data distribution $\rho$ has positive density everywhere.*

*Then for any bounded reward function $r: S \times A \to \mathbb{R}$ that lies in the span of $B$, $\pi_{z_R}$ is an optimal policy for the reward $r$.*

---

[6] Intuitively, a parametric function $f$ is overparameterized if every possible small change of $f$ can be realized by a small change of the parameter. Formally, we say that a parametric family of functions $\theta \in \Theta \mapsto f_\theta \in L^2(X, \mathbb{R}^d)$ smoothly parameterized by $\theta$, on some space $X$, is *overparameterized* if, for any $\theta$, the differential $\partial_\theta f_\theta$ is surjective from $\Theta$ to $L^2(X, \mathbb{R}^d)$. For finite $X$, this implies that the dimension of $\theta$ is larger than $\#X$. For infinite $X$, this implies that $\dim(\theta)$ is infinite, such as parameterizing functions on $[0; 1]$ by their Fourier expansion.

*More generally, for any bounded reward function $r\colon S \times A \to \mathbb{R}$, the policy $\pi_{z_R}$ is an optimal policy for the reward $r_B$ defined as the $L^2(\rho)$-orthogonal projection of $r$ onto the span of $B$.*

*Moreover,*

$$\|V^{\pi_{z_R}} - V^\star\|_\infty \leq \frac{2}{1-\gamma} \|r - r_B\|_\infty \tag{40}$$

*with $V^\star$ the optimal value function for $r$. More precisely,*

$$\|V^{\pi_{z_R}} - V^\star\|_\infty \leq \left\|(\mathrm{Id} - \gamma P_{\pi^\star})^{-1}(r - r_B)\right\|_\infty + \left\|(\mathrm{Id} - \gamma P_{\pi_{z_R}})^{-1}(r - r_B)\right\|_\infty \tag{41}$$

*with $\pi^\star$ the optimal policy for reward $r$, and notation $P_\pi$ as in Theorem 9.*

The last bounds (41) implies the previous one (40), as $(\mathrm{Id} - \gamma P_\pi)^{-1}$ is bounded by $\frac{1}{1-\gamma}$ in $L^\infty$ norm for any policy $\pi$.

**Discussion: optimal $B$ and priors on rewards.** This bound is interesting if $r - r_B$ is small, namely, if $B$ captures most of the components of the reward functions we are interested in. But even for rewards not spanned by $B$, the bound is smaller if $r - r_B$ avoids the largest eigendirections of $(\mathrm{Id} - P_\pi)^{-1}$ for various policies $\pi$, namely, if $B$ captures these largest eigendirections. These eigendirections are those of $P_\pi$: so the bound will be small if $B$ contains the largest eigendirections of $P_\pi$ for various policies $\pi$, corresponding to spatially continuity of functions under transitions in the environment ($P_\pi f$ close to $f$).

If we are interested in spatially smooth rewards $r$, then $r - r_B$ is small if $B$ if captures smooth functions first. But even for rewards not spanned by $B$, and for non-spatially smooth rewards (e.g., goal-oriented problems with reward $\mathbb{1}_{s=\text{goal}}$), the bound (41) shows that $B$ should first capture spatially smooth eigenvectors of many policies $\pi$.

Is this a natural consequence of FB training? Up to some approximations, yes. For a single $z$ and policy $\pi_z$, the loss (36) used to train $B$ is optimal when $B$ captures the largest *singular* directions of $(\mathrm{Id} - P_{\pi_z})^{-1}$, which is slightly different. The optimal policy $(\mathrm{Id} - P_{\pi^\star})^{-1}$ is not represented in the criterion, and it is not clear to what extent spatial continuity with respect to $P_{\pi_z}$ or $P_{\pi^\star}$ differ. Moreover, in the full algorithm, $B$ is shared between several $z$ and several policies. So we have no rigorous result here. Still, the intuition shows that FB training goes in the correct direction.

## B.5 Estimating $z_R$ from a Different State Distribution at Test Time

The algorithm of Section 4 computes $F$ and $B$ with respect a reference measure $\rho$ equal to the distribution $\rho$ of state-actions in the training set. Theorem 2 requires $z_R = \mathbb{E}_{(s,a)\sim\rho}[r(s,a)B(s,a)]$ to be estimated using rewards observed from the same state-action distribution $\rho$ as the one used to train $F$ and $B$.

So in general, estimating $z_R$ requires either being able to run the exploration policy again once reward samples are available, or being able to explicitly estimate $r(s,a)$ on states stored in the training set.

However, at test time, we would generally like to use the policy learned, rather than the exploration policy. This will result in a distribution of states-actions $\rho_{\text{test}}(s,a)$ different from the training distribution $\rho(s,a)$.

If the training set remains accessible, a generic solution is to train a model $\hat{r}(s,a)$ of the reward function, from rewards $r(s,a)$ observed at test time under any distribution $(s,a) \sim \rho_{\text{test}}$. Then one can estimate $z_R$ by averaging this model $\hat{r}$ over state-actions sampled from the training set:

$$\hat{z}_R := \mathbb{E}_{(s,a)\sim\rho}[\hat{r}(s,a)B(s,a)]. \tag{42}$$

However, the training set may not be available anymore at test time. But as it turns out, if we use for $\hat{r}$ a *linear* model based on the features learned in $B$, then we do not need to store the training set: it is enough to estimate the matrix $\mathrm{Cov}_\rho(B)$, which can be pre-computed during training. This is summarized in the following.

**Proposition 15.** *Let $\hat{r}$ be the linear model of rewards computed at test time by linear regression of the reward $r$ over the components $B_1, \ldots, B_d$ of $B$, with state-actions taken from a test distribution $\rho_{\text{test}}$. Explicitly,*

$$\hat{r}(s,a) := B(s,a)^\top w, \qquad w := (\mathrm{Cov}_{\rho_{\text{test}}} B)^{-1} \mathbb{E}_{(s,a)\sim\rho_{\text{test}}}[r(s,a)B(s,a)]. \tag{43}$$

*Here we assume that $\mathrm{Cov}_{\rho_{\mathrm{test}}} B = \mathbb{E}_{(s,a) \sim \rho_{\mathrm{test}}}[B(s,a)B(s,a)^\top]$ is invertible.*

*Then the estimate $\hat{z}_R$ computed by using this model $\hat{r}$ in (42) is*

$$\hat{z}_R = (\mathrm{Cov}_\rho B)(\mathrm{Cov}_{\rho_{\mathrm{test}}} B)^{-1} \mathbb{E}_{(s,a) \sim \rho_{\mathrm{test}}}[r(s,a)B(s,a)]. \tag{44}$$

*Moreover, if $\rho = \rho_{\mathrm{test}}$, or if $r$ is linear over $B$, then $\hat{z}_R = z_R$.*

For this estimate, $\mathrm{Cov}_{\rho_{\mathrm{test}}} B$ and $\mathbb{E}_{(s,a) \sim \rho_{\mathrm{test}}}[r(s,a)B(s,a)]$ can be computed at test time, while the matrix $\mathrm{Cov}_\rho B$ must be computed at training time. This way, the training set can be discarded.

If the estimate (43) is used, the learned policies correspond to using universal successor features approximators [BBQ$^+$18] on top of the features learned by $B$. Indeed, universal successor features with features $\varphi$ use the policies $\arg\max_a \psi(s,a,w)^\top w$ with $w = (\mathrm{Cov}_{\rho_{\mathrm{test}}} \varphi)^{-1} \mathbb{E}_{(s,a) \sim \rho_{\mathrm{test}}}[r(s,a)\varphi(s,a)]$ the regression vector of $r$ over the features $\varphi$, and with $\psi = \mathrm{Succ}(\varphi)$ the successor features of $\varphi$. Here we use the policies $\pi_{\hat{z}_R} = \arg\max_a F(s,a,\hat{z}_R)^\top \hat{z}_R$. Let us set $\varphi := B$ as the base features in successor features. Then the above shows that $\hat{z}_R = (\mathrm{Cov}_\rho B)w$ when using the linear model of rewards. Moreover, we proved in Theorem 13 that the optimum for $F$ is $F = (\mathrm{Cov}_\rho B)^{-1} \mathrm{Succ}(B)$. Therefore, the policies coincide in this situation.

Thus, universal successor features based on $B$ appear as a particular case if a linear model of rewards is used at test time, although in general any reward model may be used in (42).

## B.6 A Note on the Measure $M^\pi$ and its Density $m^\pi$

In finite spaces, the definition of the successor state density $m^\pi$ via

$$M^\pi(s,a,\mathrm{d}s',\mathrm{d}a') = m^\pi(s,a,s',a')\rho(\mathrm{d}s',\mathrm{d}a') \tag{45}$$

with respect to the data distribution $\rho$ poses no problem, as long as the data distribution is positive everywhere.

In continuous spaces, this can be understood as the (Radon–Nikodym) density of $M^\pi$ with respect to $\rho$, assuming $M^\pi$ has no singular part with respect to $\rho$. However, this is *never* the case: in the definition (1) of the successor state measure $M^\pi$, the term $t = 0$ produces a Dirac measure $\delta_{s,a}$. So $M^\pi$ has a singular component due to $t = 0$, and $m^\pi$ is better thought of as a distribution.

When $m^\pi$ is a distribution, a continuous parametric model $m_\theta^\pi$ learned by (5) can approximate $m^\pi$ in the weak topology only: $m_\theta^\pi \rho$ approximates $M^\pi$ for the weak convergence of measures. Thus, for the forward-backward representation, $F(s,a,z)^\top B(s',a')\rho(\mathrm{d}s',\mathrm{d}a')$ weakly approximates $M^{\pi_z}(s,a,\mathrm{d}s',\mathrm{d}a')$.

We have not found this to be a problem either in theory or practice. In particular, Theorem 8 covers weak approximations.

Alternatively, one may just define successor states starting at $t = 1$ in (1). This only works well if rewards $r(s,a)$ depend on the state $s$ but not the action $a$ (e.g., in goal-oriented settings). If starting the definition at $t = 1$, $m^\pi$ is an ordinary function provided the transition kernels $P(\mathrm{d}s'|s,a)$ of the environment are non-singular, $\rho$ has positive density, and $\pi(\mathrm{d}a|s)$ is non-singular as well. Starting at $t = 1$ induces the following changes in the theorems:

- In the learning algorithm (5) for successor states, the term $\partial_\theta m_\theta^\pi(s,a,s,a)$ becomes $\gamma \, \partial_\theta m_\theta^\pi(s,a,s',a')$.
- The expression for the $Q$-function in Theorem 2 becomes $Q^\star(s,a) = r(s,a) + F(s,a,z_R)^\top z_R$, and likewise in Theorem 4. The $r(s,a)$ term covers the immediate reward at a state, since we have excluded $t = 0$ from the definition of successor states.
- In general the expression for optimal policies becomes

$$\pi_z(s) := \arg\max_a \{r(s,a) + F(s,a,z)^\top z\} \tag{46}$$

  which cannot be computed from $z$ and $F$ alone in the unsupervised training phase. The algorithm only makes sense for rewards that depend on $s$ but not on $a$ (e.g., in goal-oriented settings): then the policy $\pi_z$ is equal to $\pi_z(s) := \arg\max_a F(s,a,z)^\top z$ again.

## C Proofs

The first proposition is a direct consequence of the definition (10) of successor states with a goal space $G$.

**Proposition 16.** *Let $\varphi\colon S \times A \to G$ be a map to some goal space $G$.*

*Let $\pi$ be some policy, and let $M^\pi$ be the successor state measure (10) of $\pi$ in goal space $G$. Let $m^\pi$ be the density of $M^\pi$ with respect to some positive measure $\rho$ on $G$.*

*Let $r\colon G \to \mathbb{R}$ be some function on $G$, and define the reward function $R(s,a) := r(\varphi(s,a))$ on $S \times A$.*

*Then the $Q$-function $Q^\pi$ of policy $\pi$ for reward $R$ is*

$$Q^\pi(s,a) = \int r(g)\, M^\pi(s,a,\mathrm{d}g) \tag{47}$$

$$= \int_{g \in G} r(g)\, m^\pi(s,a,g)\, \rho(\mathrm{d}g). \tag{48}$$

*Proof of Proposition 16.* For each time $t \geq 0$, let $P_t^\pi(s_0,a_0,\mathrm{d}g)$ be the probability distribution of $g = \varphi(s_t,a_t)$ over trajectories of the policy $\pi$ starting at $(s_0,a_0)$ in the MDP. Thus, by the definition (10),

$$M^\pi(s,a,\mathrm{d}g) = \sum_{t \geq 0} \gamma^t P_t^\pi(s,a,\mathrm{d}g). \tag{49}$$

The $Q$-function of $\pi$ for the reward $R$ is by definition (the sums and integrals are finite since $R$ is bounded)

$$Q^\pi(s,a) = \sum_{t \geq 0} \gamma^t\, \mathbb{E}[R(s_t,a_t) \mid s_0 = s,\, a_0 = a,\, \pi] \tag{50}$$

$$= \sum_{t \geq 0} \gamma^t\, \mathbb{E}[r(\varphi(s_t,a_t)) \mid s_0 = s,\, a_0 = a,\, \pi] \tag{51}$$

$$= \sum_{t \geq 0} \gamma^t \int_g r(g) P_t^\pi(s,a,\mathrm{d}g) \tag{52}$$

$$= \int_g r(g) M^\pi(s,a,\mathrm{d}g) \tag{53}$$

$$= \int_g r(g) m^\pi(s,a,g)\rho(\mathrm{d}g) \tag{54}$$

by definition of the density $m^\pi$. $\qquad\square$

*Proof of Theorems 2 and 4.* Theorem 2 is a particular case of Theorem 4 ($\varphi = \mathrm{Id}$ and $\bar{m} = 0$), so we only prove the latter.

Let $R(s,a) = r(\varphi(s,a))$ be a reward function as in the theorem.

For any policy $\pi$, let $M^\pi$ be its successor measure defined by (10), and let $m^\pi$ denote its density with respect to $\rho$.

The $Q$-function of $\pi$ for the reward $R$ is, by Proposition 16,

$$Q^\pi(s,a) = \int_g r(g) m^\pi(s,a,g)\rho(\mathrm{d}g) \tag{55}$$

The definition of an extended FB representation states that for any $z \in Z$, $m^{\pi_z}(s,a,g)$ is equal to $F(s,a,z)^\top B(g) + \bar{m}(s,z,g)$.

Therefore, for any $z \in Z$ we have

$$Q^{\pi_z}(s,a) = \int_g r(g) \left( F(s,a,z)^\top B(g) + \bar{m}(s,z,g) \right) \rho(\mathrm{d}g) \tag{56}$$

$$= F(s,a,z)^\top \int_g r(g) B(g) \rho(\mathrm{d}g) + \int_g r(g) \bar{m}(s,z,g) \rho(\mathrm{d}g) \tag{57}$$

$$= F(s,a,z)^\top z_R + \bar{V}^z(s) \tag{58}$$

by definition of $z_R$ and $\bar{V}$. This proves the claim (15) about $Q$-functions.

By definition, the policy $\pi_z$ selects the action $a$ that maximizes $F(s,a,z)^\top z$. Take $z = z_R$. Then

$$\pi_{z_R} = \arg\max_a F(s,a,z_R)^\top z_R \tag{59}$$

$$= \arg\max_a \left\{ F(s,a,z_R)^\top z_R + \bar{V}^z(s) \right\} \tag{60}$$

since the last term does not depend on $a$.

This quantity is equal to $Q^{\pi_{z_R}}(s,a)$. Therefore,

$$\pi_{z_R} = \arg\max_a Q^{\pi_{z_R}}(s,a) \tag{61}$$

and by the above, $Q^{\pi_{z_R}}(s,a)$ is indeed equal to the $Q$-function of policy $\pi_{z_R}$ for the reward $R$. Therefore, $\pi_{z_R}$ and $Q^{\pi_{z_R}}$ constitute an optimal Bellman pair for reward $R$. Since $Q^{\pi_{z_R}}(s,a)$ is the $Q$-function of $\pi_{z_R}$, it satisfies the Bellman equation

$$Q^{\pi_{z_R}}(s,a) = R(s,a) + \gamma \, \mathbb{E}_{s'|(s,a)} \, Q^{\pi_{z_R}}(s', \pi_{z_R}(s')) \tag{62}$$

$$= R(s,a) + \gamma \, \mathbb{E}_{s'|(s,a)} \max_{a'} Q^{\pi_{z_R}}(s', a') \tag{63}$$

by (61). This is the optimal Bellman equation for $R$, and $\pi_{z_R}$ is the optimal policy for $R$.

We still have to prove the last statement of Theorem 4. Since $\pi_{z_R}$ is an optimal policy for $R$, for any other policy $\pi_z$ and state-action $(s,a)$ we have

$$Q^{\pi_{z_R}}(s,a) \geq Q^{\pi_z}(s,a). \tag{64}$$

Using the formulas above for $Q^\pi$, with $\bar{m} = 0$, this rewrites as

$$F(s,a,z_R)^\top z_R \geq F(s,a,z)^\top z_R \tag{65}$$

as needed. Thus $F(s,a,z_R)^\top z_R \geq \sup_{z \in Z} F(s,a,z)^\top z_R$, and equality occurs by taking $z = z_R$. This ends the proof of Theorem 4. $\qquad\square$

*Proof of Proposition 6.* Assume $d = \#S \times \#A$; extra dimensions can just be ignored by setting the extra components of $F$ and $B$ to 0.

With $d = \#S \times \#A$, we can index the components of $Z$ by pairs $(s,a)$.

First, let us set $B(s,a) := \mathbb{1}_{s,a}$.

Let $r \colon S \times A \to \mathbb{R}$ be any reward function. Let $z_R \in \mathbb{R}^{\#S \times \#A}$ be defined as in Theorem 2, namely,

$$z_R = \sum_{(s,a)} r(s,a) B(s,a) \rho(s,a). \tag{66}$$

With our choice of $B$, the components of $z_R$ are $(z_R)_{s,a} = r(s,a)\rho(s,a)$. Since $\rho > 0$, the correspondence $r \leftrightarrow z_R$ is bijective.

Let us now define $F$. Take $z \in Z$. Since $r \leftrightarrow z_R$ is bijective, this $z$ is equal to $z_R$ for some reward function $r$. Let $\pi_z$ be an optimal policy for this reward $r$ in the MDP. Let $M^{\pi_r}$ be the successor state measure of policy $\pi_z$, namely:

$$M^{\pi_z}(s,a,s',a') = \sum_{t \geq 0} \gamma^t \Pr\left( (s_t, a_t) = (s', a') \mid (s_0, a_0) = (s,a), \pi_z \right). \tag{67}$$

Now define $F(s, a, z)$ by setting its $(s', a')$ component to $M^{\pi_z}(s, a, s', a')/\rho(s', a')$ for each $(s', a')$:

$$F(s, a, z)_{s', a'} := M^{\pi_z}(s, a, s', a')/\rho(s', a'). \tag{68}$$

Then we have

$$F(s, a, z)^\top B(s', a') = \sum_{s'', a''} F(s, a, z)_{s'', a''} B(s', a')_{s'', a''}$$

$$= F(s, a, z)_{s', a'} = M^{\pi_z}(s, a, s', a')/\rho(s', a') \tag{69}$$

because by our choice of $B$, $B(s', a')_{s'' a''} = \mathbb{1}_{s'=s'', \, a'=a''}$.

Thus, $F(s, a, z)^\top B(s', a')$ is the density of the successor state measure $M^{\pi_z}$ of policy $\pi_z$ with respect to $\rho$, as needed.

We still have to check that $\pi_z$ satisfies $\pi_z(s) = \arg\max F(s, a, z)^\top z$ (since this is not how it was defined). Since $\pi_z$ was defined as an optimal policy for the reward $r$ associated with $z$, it satisfies

$$\pi_z(s) = \arg\max_a Q^{\pi_z}(s, a) \tag{70}$$

with $Q^{\pi_z}(s, a)$ the $Q$-function of policy $\pi_z$ for the reward $r$. This $Q$-function is equal to the cumulated expected reward

$$Q^{\pi_z}(s, a) = \sum_{t \geq 0} \gamma^t \, \mathbb{E}\left[r(s_t, a_t) \mid s_0 = s, \, a_0 = a, \, \pi_z\right] \tag{71}$$

$$= \sum_{t \geq 0} \gamma^t \sum_{s', a'} r(s', a') \Pr\left((s_t, a_t) = (s', a') \mid s_0 = s, \, a_0 = a, \, \pi_z\right) \tag{72}$$

$$= \sum_{s', a'} r(s', a') \sum_{t \geq 0} \gamma^t \Pr\left((s_t, a_t) = (s', a') \mid s_0 = s, \, a_0 = a, \, \pi_z\right) \tag{73}$$

$$= \sum_{s', a'} r(s', a') M^{\pi_z}(s, a, s', a') \tag{74}$$

$$= \sum_{s', a'} r(s', a') F(s, a, z)_{s', a'} \, \rho(s', a') \tag{75}$$

$$= F(s, a, z)^\top \left(\sum_{s' a'} r(s', a') \rho(s', a') \mathbb{1}_{s' a'}\right) \tag{76}$$

$$= F(s, a, z)^\top z \tag{77}$$

since $z$ is equal to $\sum_{(s', a')} r(s', a') B(s', a') \rho(s', a')$. This proves that $\pi_z(s) = \arg\max_a Q^{\pi_z(s, a)} = \arg\max_a F(s, a, z)^\top z$. So this choice of $F$ and $B$ satisfies all the properties claimed. $\qquad\square$

We will rely on the following two basic results in $Q$-learning.

**Proposition 17** ($r \mapsto Q^\star$ is Lipschitz in sup-norm). *Let $r_1$, $r_2 \colon S \times A \to \mathbb{R}$ be two bounded reward functions. Let $Q_1^\star$ and $Q_2^\star$ be the corresponding optimal $Q$-functions, and likewise for the $V$-functions. Then*

$$\sup_{S \times A} |Q_1^\star - Q_2^\star| \leq \frac{1}{1 - \gamma} \sup_{S \times A} |r_1 - r_2| \text{ and } \sup_S |V_1^\star - V_2^\star| \leq \frac{1}{1 - \gamma} \sup_{S \times A} |r_1 - r_2|. \tag{78}$$

*Moreover for any policy $\pi$, we have*

$$\sup_{S \times A} |Q_1^\pi - Q_2^\pi| \leq \frac{1}{1 - \gamma} \sup_{S \times A} |r_1 - r_2| \text{ and } \sup_S |V_1^\pi - V_2^\pi| \leq \frac{1}{1 - \gamma} \sup_{S \times A} |r_1 - r_2|. \tag{79}$$

*Proof.* Assume $\sup_{S \times A} |r_1 - r_2| \leq \varepsilon$ for some $\varepsilon \geq 0$.

For any policy $\pi$, let $Q_1^\pi$ be its $Q$-function for reward $r_1$, and likewise for $r_2$. Let $\pi_1$ and $\pi_2$ be optimal policies for $r_1$ and $r_2$, respectively. Then for any $(s,a) \in S \times A$,

$$Q_1^\star(s,a) = Q_1^{\pi_1}(s,a) \tag{80}$$

$$\geq Q_1^{\pi_2}(s,a) \tag{81}$$

$$= \sum_{t \geq 0} \gamma^t \, \mathbb{E}\left[r_1(s_t, a_t) \mid \pi_2, (s_0, a_0) = (s,a)\right] \tag{82}$$

$$\geq \sum_{t \geq 0} \gamma^t \, \mathbb{E}\left[r_2(s_t, a_t) - \varepsilon \mid \pi_2, (s_0, a_0) = (s,a)\right] \tag{83}$$

$$= \sum_{t \geq 0} \gamma^t \, \mathbb{E}\left[r_2(s_t, a_t) \mid \pi_2, (s_0, a_0) = (s,a)\right] - \frac{\varepsilon}{1 - \gamma} \tag{84}$$

$$= Q_2^\star(s,a) - \frac{\varepsilon}{1 - \gamma} \tag{85}$$

and likewise in the other direction, which ends the proof for $Q$-functions. The case of $V$-functions follows by restricting to the optimal actions at each state $s$.

Now, let $\pi$ a policy. We have

$$|Q_1^\pi(s,a) - Q_2^\pi(s,a)| = \left| \sum_{t \geq 0} \gamma^t \, \mathbb{E}\left[r_1(s_t, a_t) \mid \pi, (s_0, a_0) = (s,a)\right] - \sum_{t \geq 0} \gamma^t \, \mathbb{E}\left[r_2(s_t, a_t) \mid \pi, (s_0, a_0) = (s,a)\right] \right|$$
$$\tag{86}$$

$$\leq \sum_{t \geq 0} \gamma^t \, \mathbb{E}\left[|r_2(s_t, a_t) - r_1(s_t, a_t)| \mid \pi, (s_0, a_0) = (s,a)\right] \tag{87}$$

$$\leq \frac{1}{1 - \gamma} \sup_{S \times A} |r_1 - r_2| . \tag{88}$$

The case of $V$-functions follows by taking the expectation over actions according to $\pi$. $\qquad \square$

**Proposition 18.** *Let $f \colon S \times A \to \mathbb{R}$ be any function, and define a policy $\pi_f$ by $\pi_f(s) := \arg\max_a f(s,a)$. Let $r \colon S \times A \to \mathbb{R}$ be some bounded reward function. Let $Q^\star$ be its optimal $Q$-function, and let $Q^{\pi_f}$ be the $Q$-function of $\pi_f$ for reward $r$.*

*Then*

$$\sup_{S \times A} |f - Q^\star| \leq \frac{2}{1 - \gamma} \sup_{S \times A} |f - Q^{\pi_f}| \tag{89}$$

*and*

$$\sup_{S \times A} |Q^{\pi_f} - Q^\star| \leq \frac{3}{1 - \gamma} \sup_{S \times A} |f - Q^{\pi_f}| \tag{90}$$

*Proof.* Define $\varepsilon(s,a) := Q^{\pi_f}(s,a) - f(s,a)$.

The $Q$-function $Q^{\pi_f}$ satisfies the Bellman equation

$$Q^{\pi_f}(s,a) = r(s,a) + \gamma \, \mathbb{E}_{s'|(s,a)} Q^{\pi_f}(s', \pi_f(s')) \tag{91}$$

for any $(s,a) \in S \times A$. Substituting $Q^{\pi_f} = f + \varepsilon$, this rewrites as

$$f(s,a) = r(s,a) - \varepsilon(s,a) + \gamma \, \mathbb{E}_{s'|(s,a)} \left[f(s', \pi_f(s')) + \varepsilon(s', \pi_f(s'))\right] \tag{92}$$

$$= r(s,a) - \varepsilon'(s,a) + \gamma \, \mathbb{E}_{s'|(s,a)} f(s', \pi_f(s')) \tag{93}$$

$$= r(s,a) - \varepsilon'(s,a) + \gamma \, \mathbb{E}_{s'|(s,a)} \max_{a'} f(s', a') \tag{94}$$

by definition of $\pi_f$, where we have set

$$\varepsilon'(s,a) := \varepsilon(s,a) - \gamma \, \mathbb{E}_{s'|(s,a)} \varepsilon(s', \pi_f(s')). \tag{95}$$

(94) is the optimal Bellman equation for the reward $r - \varepsilon'$. Therefore, $f$ is the optimal $Q$-function for the reward $r - \varepsilon'$. Since $Q^\star$ is the optimal $Q$-function for reward $r$, by Proposition 17, we have

$$\sup_{S \times A} |f - Q^\star| \leq \frac{1}{1 - \gamma} \sup_{S \times A} |\varepsilon'| \tag{96}$$

By construction of $\varepsilon'$, $\sup_{S \times A} |\varepsilon'| \leq 2 \sup_{S \times A} |\varepsilon| = 2 \sup_{S \times A} |f - Q^{\pi_f}|$. This proves the first claim.

The second claim follows by the triangle inequality $|Q^{\pi_f} - Q^\star| \leq |Q^{\pi_f} - f| + |f - Q^\star|$ and $\frac{2}{1-\gamma} + 1 \leq \frac{3}{1-\gamma}$. $\qquad\square$

*Proof of Theorem 8.* By construction of the Kantorovich–Rubinstein norm, the second claim of Theorem 8 is a particular case of the third claim, with $\|f\|_A := \max(\|f\|_\infty, \|f\|_{\mathrm{Lip}})$ and $\|\mu\|_B := \|\mu\|_{\mathrm{KR}}$.

Likewise, since $m$ is the density of $M$ with respect to $\rho$, the first claim is an instance of the third, by taking $\|f\|_A := \|f\|_\infty$ and $\|\mu\|_B := \left\| \frac{\mathrm{d}\mu}{\mathrm{d}\rho} \right\|_{L^1(\rho)}$. Therefore, we only prove the third claim.

Let $z \in Z$ and let $r \colon S \times A \to \mathbb{R}$ be any reward function. By Proposition 16 with $G = S \times A$ and $\varphi = \mathrm{Id}$, the $Q$-function of policy $Q^{\pi_z}$ for this reward is

$$Q^{\pi_z}(s, a) = \int r(s', a') \, M^{\pi_z}(s, a, \mathrm{d}s', \mathrm{d}a'). \tag{97}$$

Let $\varepsilon_z(s, a, \mathrm{d}s', \mathrm{d}a')$ be the difference of measures between the model $F^\top B \rho$ and $M^{\pi_z}$:

$$\varepsilon_z(s, a, \mathrm{d}s', \mathrm{d}a') := M^{\pi_z}(s, a, \mathrm{d}s', \mathrm{d}a') - \hat{M}^z(s, a, \mathrm{d}s', \mathrm{d}a') \tag{98}$$

$$= M^{\pi_z}(s, a, \mathrm{d}s', \mathrm{d}a') - F(s, a, z)^\top B(s', a') \rho(\mathrm{d}s', \mathrm{d}a'). \tag{99}$$

We want to control the optimality gap in terms of $\sup_{s,a} \|\varepsilon_z(s, a, \cdot)\|_B$.

By definition of $\varepsilon_z$,

$$Q^{\pi_z}(s, a) = \int r(s', a') F(s, a, z)^\top B(s', a') \rho(\mathrm{d}s', \mathrm{d}a') + \int r(s', a') \, \varepsilon_z(s, a, \mathrm{d}s', \mathrm{d}a') \tag{100}$$

$$= F(s, a, z)^\top z_R + \int r(s', a') \, \varepsilon_z(s, a, \mathrm{d}s', \mathrm{d}a') \tag{101}$$

since $z_R = \int r(s', a') B(s', a') \rho(\mathrm{d}s', \mathrm{d}a')$. Therefore,

$$\left| Q^{\pi_z}(s, a) - F(s, a, z)^\top z_R \right| = \left| \int r(s', a') \, \varepsilon_z(s, a, \mathrm{d}s', \mathrm{d}a') \right| \tag{102}$$

$$\leq \|r\|_A \, \|\varepsilon_z(s, a, \cdot)\|_B \tag{103}$$

for any reward $r$ and any $z \in Z$ (not necessarily $z = z_R$).

Let $Q^\star$ be the optimal $Q$-function for reward $r$. Define $f(s, a) := F(s, a, z_R)^\top z_R$. By definition, the policy $\pi_{z_R}$ is equal to $\arg\max_a f(s, a)$. Therefore, by Proposition 18,

$$\sup_{S \times A} |Q^{\pi_{z_R}} - Q^\star| \leq \frac{3}{1 - \gamma} \sup_{S \times A} |f - Q^{\pi_{z_R}}|. \tag{104}$$

and

$$\sup_{S \times A} |f - Q^\star| \leq \frac{2}{1 - \gamma} \sup_{S \times A} |f - Q^{\pi_{z_R}}|. \tag{105}$$

But by the above,

$$\sup_{S \times A} |f - Q^{\pi_{z_R}}| = \sup_{S \times A} \left| F(s, a, z_R)^\top z_R - Q^{\pi_{z_R}}(s, a) \right| \tag{106}$$

$$\leq \|r\|_A \sup_{S \times A} \|\varepsilon_{z_R}(s, a, \cdot)\|_B. \tag{107}$$

Therefore, for any reward function $r$,

$$\sup_{S \times A} |Q^{\pi_{z_R}} - Q^\star| \leq \frac{3 \|r\|_A}{1 - \gamma} \sup_{S \times A} \|\varepsilon_{z_R}(s, a, \cdot)\|_B. \tag{108}$$

This inequality transfers to the value functions, hence the result. In addition, using again $f(s, a) = F(s, a, z_R)^\top z_R$, we obtain

$$\sup_{S \times A} \left| F(s, a, z_R)^\top z_R - Q^\star(s, a) \right| \leq \frac{2 \|r\|_A}{1 - \gamma} \sup_{S \times A} \|\varepsilon_{z_R}(s, a, \cdot)\|_B . \tag{109}$$

$\square$

**Proposition 19** (Pointwise optimality gap). *Assume the state space is finite, and view rewards and Q-functions as vectors over state-actions.*

*Let $r_1$ and $r_2$ be two reward functions, and let $\pi_1$ and $\pi_2$ be optimal policies for $r_1$ and $r_2$ respectively. Let $P_1$ and $P_2$ be the stochastic transition matrices over state-actions induced by $\pi_1$ and $\pi_2$.*

*Then the optimality gap of policy $\pi_2$ on reward $r_1$ is at most*

$$0 \leq Q_{r_1}^{\pi_1} - Q_{r_1}^{\pi_2} \leq \left( (\mathrm{Id} - \gamma P_1)^{-1} - (\mathrm{Id} - \gamma P_2)^{-1} \right) (r_1 - r_2) \tag{110}$$

*where the equality holds componentwise viewing the Q-functions as vectors over state-actions.*

*Proof.* This is a classical result. The inequality $0 \leq Q_{r_1}^{\pi_1} - Q_{r_1}^{\pi_2}$ is trivial since $\pi_1$ is optimal for $r_1$ and the optimal policy is optimal at every state-action. Denote $M_1 := (\mathrm{Id} - \gamma P_1)^{-1}$ and likewise for $M_2$. Then for any reward function $r$ one has $Q_r^{\pi_1} = M_1 r$ and likewise for $M_2$. Therefore

$$Q_{r_1}^{\pi_1} - Q_{r_1}^{\pi_2} = M_1 r_1 - M_2 r_1 \tag{111}$$
$$= M_1 r_1 - M_1 r_2 + M_1 r_2 - M_2 r_1 \tag{112}$$
$$\leq M_1 r_1 - M_1 r_2 + M_2 r_2 - M_2 r_1 \qquad \text{since } \pi_2 \text{ is optimal for } r_2 \tag{113}$$
$$= (M_1 - M_2)(r_1 - r_2). \tag{114}$$

$\square$

*Proof of Theorem 9.* Let $f \colon S \times A \to \mathbb{R}$ be any function. Define the policy $\pi_f(s) := \arg\max_a f(s, a)$. Define $r' := (\mathrm{Id} - \gamma P_{\pi_f})f$. The equality $f = r' + \gamma P_{\pi_f} f$ can be rewritten as

$$f(s, a) = r'(s, a) + \gamma \mathbb{E}_{s' \sim P(\mathrm{d}s'|s, a)} f(s', \pi_f(s')). \tag{115}$$

But by definition of $\pi_f$, $\pi_f(s') = \arg\max_{a'} f(s', a')$. Therefore

$$f(s, a) = r'(s, a) + \gamma \mathbb{E}_{s' \sim P(\mathrm{d}s'|s, a)} \max_{a'} f(s', a') \tag{116}$$

namely, $f$ is the optimal Q-function for reward $r'$, with $\pi_f$ the corresponding optimal policy.

Let $r$ be any reward function and let $Q^{\pi_f}$ be the Q-function of $\pi_f$ for $r$. By definition, it satisfies the Bellman equation $Q^{\pi_f} = r + \gamma P_{\pi_f} Q^{\pi_f}$, namely, $r = (\mathrm{Id} - \gamma P_{\pi_f})Q^{\pi_f}$. Therefore,

$$r - r' = (\mathrm{Id} - \gamma P_{\pi_f})(Q^{\pi_f} - f). \tag{117}$$

For any policy $\pi$, denote $M_\pi := (\mathrm{Id} - \gamma P_\pi)^{-1}$. ($M_\pi$ is the successor measure $M^\pi$ seen as a matrix.) The Q-function of $\pi$ for reward $r$ is $M_\pi r$.

Since $\pi^\star$ is optimal for $r$, and $\pi_f$ is optimal for $r'$, Proposition 19 yields

$$0 \leq Q^\star - Q^{\pi_f} \leq (M_{\pi^\star} - M_{\pi_f})(r - r') \tag{118}$$
$$= (M_{\pi^\star} - M_{\pi_f})(\mathrm{Id} - \gamma P_{\pi_f})(Q^{\pi_f} - f). \tag{119}$$

Since $M_{\pi_f}$ is the inverse of $\mathrm{Id} - \gamma P_{\pi_f}$ and likewise for $\pi^\star$, we have

$$(M_{\pi^\star} - M_{\pi_f})(\mathrm{Id} - \gamma P_{\pi_f}) = M_{\pi^\star}(\mathrm{Id} - \gamma P_{\pi_f}) - \mathrm{Id} \tag{120}$$
$$= M_{\pi^\star}(\mathrm{Id} - \gamma P_{\pi_f} - (\mathrm{Id} - \gamma P_{\pi^\star})) \tag{121}$$
$$= \gamma M_{\pi^\star}(P_{\pi^\star} - P_{\pi_f}) \tag{122}$$

and therefore

$$0 \leq Q^\star - Q^{\pi_f} \leq \gamma M_{\pi^\star}(P_{\pi^\star} - P_{\pi_f})(Q^{\pi_f} - f). \tag{123}$$

Now set

$$f(s,a) := F(s,a,z_R)^\top z_R \tag{124}$$

or in matrix notation, $f = F(z_R)^\top z_R$. Then $\pi_f = \pi_{z_R}$ by definition. Thus $Q^{\pi_f} = M_{\pi_{z_R}} r = m^{\pi_{z_R}} \operatorname{diag}(\rho)r$ in matrix notation. Moreover, $z_R = \mathbb{E}_\rho[B(s,a)r(s,a)] = B\operatorname{diag}(\rho)r$ in matrix notation. So $f = F(z_R)^\top B \operatorname{diag}(\rho)r$. Therefore,

$$Q^{\pi_f} - f = (m^{\pi_{z_R}} - F(z_R)^\top B)\operatorname{diag}(\rho)r \tag{125}$$

and

$$Q^\star - Q^{\pi_f} \le \gamma M^\star(P_{\pi^\star} - P_{\pi_f})\left(m^{\pi_{z_R}} - F(z_R)^\top B\right)\operatorname{diag}(\rho)r \tag{126}$$

and using $M_{\pi^\star} = \sum_{t\ge 0}\gamma^t P_{\pi^\star}^t$ provides the required inequality.

As a remark, the same proof works on continuous state spaces, by viewing all matrices as linear operators over functions on $S \times A$. $\qquad\square$

*Proof of Proposition 10.* This is just a triangle inequality.

$$\|V_r^{\pi_{\hat z_R}} - V_r^\star\|_\infty \le \|V_r^{\pi_{\hat z_R}} - V_{\hat r}^{\pi_{\hat z_R}}\|_\infty + \|V_{\hat r}^{\pi_{\hat z_R}} - V_{\hat r}^\star\|_\infty + \|V_{\hat r}^\star - V_r^\star\|_\infty \tag{127}$$

$$\le \frac{\|r - \hat r\|_\infty}{1-\gamma} + \varepsilon_{FB} + \frac{\|r - \hat r\|_\infty}{1-\gamma}. \tag{128}$$

The last inequality follows from two facts: by assumption, the difference between the value function of $\pi_{\hat z_R}$ and the optimal value function $V_{\hat r}^\star$ is at most $\varepsilon_{FB}$, then by Proposition 17, the difference between $V_{\hat r}^\star$ and $V_r^\star$ as well as the difference between $V_r^{\pi_{\hat z_R}}$ and $V_{\hat r}^{\pi_{\hat z_R}}$ are bounded by $\frac{1}{1-\gamma}\sup_{S\times A}|\hat r - r|$. $\qquad\square$

*Proof of Theorem 12.* We proceed by building a reward function $\hat r$ corresponding to $\hat z_R$. Then we will bound $\hat r - r$ and apply Proposition 10.

First, by Remark 7, up to reducing $d$, we can assume that $B$ is $L^2(\rho)$-orthonormal.

For any function $\varphi\colon (s,a) \to \mathbb{R}$, define $z_\varphi := \mathbb{E}_{(s,a)\sim\rho}[\varphi(s,a)B(s,a)]$. For each $z \in Z$, define $\varphi_z$ via $\varphi_z(s,a) := B(s,a)^\top z$. Then, if $B$ is $L^2(\rho)$-orthonormal, we have $z_{\varphi_z} = z$. (Indeed, $z_{\varphi_z} = \mathbb{E}[(B(s,a)^\top z)B(s,a)] = \mathbb{E}[B(s,a)(B(s,a)^\top z)] = \left(\mathbb{E}[B(s,a)B(s,a)^\top]\right)z$.)

Define the function

$$\hat r := r + \varphi_{\hat z_R - z_R} \tag{129}$$

using the functions $\varphi_z$ defined above. By construction, $z_{\hat r} = z_R + z_{\varphi_{\hat z_R - z_R}} = \hat z_R$. Therefore, the policy $\pi_{\hat z_R}$ associated to $\hat z_R$ is the policy associated to the reward $\hat r$.

We will now apply Proposition 10 to $r$ and $\hat r$. For this, we need to bound $\|\varphi_{\hat z_R - z_R}\|_\infty$.

Let $B_1, B_2, \ldots, B_d$ be the components of $B$ as functions on $S \times A$. For any $z \in Z$, we have

$$\|\varphi_z\|_{L^2(\rho)}^2 = \left\|\sum_i z_i B_i\right\|_{L^2(\rho)}^2 = \sum_i z_i^2 = \|z\|^2 \tag{130}$$

since the $B_i$ are $L^2(\rho)$-orthonormal. Moreover, by construction, $\varphi_z$ lies in the linear span $\langle B\rangle$ of the functions $(B_i)$. Therefore

$$\|\varphi_z\|_\infty \le \zeta(B)\|\varphi_z\|_{L^2(\rho)} = \zeta(B)\|z\| \tag{131}$$

by the definition of $\zeta(B)$ (Definition 11).

Therefore,

$$\|\varphi_{\hat z_R - z_R}\|_\infty \le \zeta(B)\|\hat z_R - z_R\|. \tag{132}$$

Let us now bound $\hat{z}_R - z_R$:

$$\mathbb{E}\left[\|\hat{z}_R - z_R\|^2\right] = \mathbb{E}\left[\mathbb{E}\left[\|\hat{z}_R - z_R\|^2 \mid (s_i, a_i)\right]\right] \tag{133}$$

$$= \mathbb{E}\left[\mathbb{E}\left[\|\hat{z}_R - \mathbb{E}[\hat{z}_R \mid (s_i, a_i)]\|^2 + \|\mathbb{E}[\hat{z}_R \mid (s_i, a_i)] - z_R\|^2 \mid (s_i, a_i)\right]\right] \tag{134}$$

$$= \mathbb{E}\left[\mathbb{E}\left[\left\|\frac{1}{N}\sum_i (\hat{r}_i - r(s_i, a_i))B(s_i, a_i)\right\|^2 + \left\|\frac{1}{N}\sum_i r(s_i, a_i)B(s_i, a_i) - z_R\right\|^2 \mid (s_i, a_i)\right]\right] \tag{135}$$

The first term satisfies

$$\mathbb{E}\left[\left\|\frac{1}{N}\sum_i (\hat{r}_i - r(s_i, a_i))B(s_i, a_i)\right\|^2 \mid (s_i, a_i)\right] = \frac{1}{N^2}\sum_i \mathbb{E}\left[(\hat{r}_i - r(s_i, a_i))^2 \|B(s_i, a_i)\|^2\right] \tag{136}$$

$$\leq \frac{1}{N^2}\sum_i v \|B(s_i, a_i)\|^2 \tag{137}$$

because the $\hat{r}_i$ are independent conditionally to $(s_i, a_i)$, and because $B$ is deterministic. The expectation of this over $(s_i, a_i)$ is

$$\mathbb{E}\left[\frac{1}{N^2}\sum_i v \|B(s_i, a_i)\|^2\right] = \frac{v}{N}\mathbb{E}_{(s,a)\sim\rho}\|B(s, a)\|^2 = \frac{v}{N}\|B\|^2_{L^2(\rho)} \tag{138}$$

which is thus a bound on the first term.

The second term satisfies

$$\mathbb{E}\left[\left\|\frac{1}{N}\sum_i r(s_i, a_i)B(s_i, a_i) - z_R\right\|^2\right] = \frac{1}{N}\|r(s,a)B(s,a) - \mathbb{E}_\rho[r(s,a)B(s,a)]\|^2_{L^2(\rho)} \tag{139}$$

since the $(s_i, a_i)$ are independent with distribution $\rho$. By the Cauchy–Schwarz inequality (applied to each component of $B$), this is at most

$$\frac{1}{N}\|r(s,a) - \mathbb{E}_\rho r\|^2_{L^2(\rho)} \|B\|^2_{L^2(\rho)}. \tag{140}$$

Therefore,

$$\mathbb{E}\left[\|\hat{z}_R - z_R\|^2\right] \leq \left(v + \|r(s,a) - \mathbb{E}_\rho r\|^2_{L^2(\rho)}\right)\frac{\|B\|^2_{L^2(\rho)}}{N}. \tag{141}$$

Since $B$ is orthonormal in $L^2(\rho)$, we have $\|B\|^2_{L^2(\rho)} = d$. Putting everything together, we find

$$\mathbb{E}\left[\|\hat{r} - r\|^2_\infty\right] \leq \frac{\zeta(B)\,d}{N}\left(v + \|r(s,a) - \mathbb{E}_\rho r\|^2_{L^2(\rho)}\right). \tag{142}$$

Therefore, by the Markov inequality, for any $\delta > 0$, with probability $1 - \delta$,

$$\|\hat{r} - r\|_\infty \leq \sqrt{\frac{\zeta(B)\,d}{N\delta}\left(v + \|r(s,a) - \mathbb{E}_\rho r\|^2_{L^2(\rho)}\right)} \tag{143}$$

hence the conclusion by Proposition 10. $\qquad\square$

*Proof of Theorem 13.* Let

$$m(s, a, s', a') := \sum_{t\geq 0}\gamma^t \frac{P_t(\mathrm{d}s', \mathrm{d}a'\mid s, a, \pi_z)}{\rho(\mathrm{d}s', \mathrm{d}a')} \tag{144}$$

so that

$$\ell(F, B) = \int \left| F(s, a, z)^\top B(s', a') - m(s, a, s', a') \right|^2 \rho(\mathrm{d}s, \mathrm{d}a)\rho(\mathrm{d}s', \mathrm{d}a'). \qquad (145)$$

Let us first take the derivative with respect to the parameters of $F$. This is 0 by assumption, so we find

$$0 = \int \partial_\theta F(s, a, z)^\top B(s', a') \left( F(s, a, z)^\top B(s', a') - m(s, a, s', a') \right) \rho(\mathrm{d}s, \mathrm{d}a)\rho(\mathrm{d}s', \mathrm{d}a') \quad (146)$$

$$= \int \partial_\theta F(s, a, z)^\top G(s, a)\rho(\mathrm{d}s, \mathrm{d}a) \qquad (147)$$

where

$$G(s, a) := \int B(s', a') \left( F(s, a, z)^\top B(s', a') - m(s, a, s', a') \right) \rho(\mathrm{d}s', \mathrm{d}a') \qquad (148)$$

Since the model is overparameterized, we can realize any $L^2$ function $f(s, a)$ as the derivative $\partial_\theta F(s, a, z)$ for some direction $\theta$. Therefore, the equation $0 = \int \partial_\theta F(s, a, z)^\top G(s, a)\rho(\mathrm{d}s, \mathrm{d}a)$ implies that $G(s, a)$ is $L^2(\rho)$-orthogonal to any function $f(s, a)$ in $L^2(\rho)$. Therefore, $G(s, a)$ vanishes $\rho$-almost everywhere, namely

$$\int B(s', a')F(s, a, z)^\top B(s', a')\rho(\mathrm{d}s', \mathrm{d}a') = \int B(s', a')m(s, a, s', a')\rho(\mathrm{d}s', \mathrm{d}a') \qquad (149)$$

Now, since $F(s, a, z)^\top B(s', a')$ is a real number, $F(s, a, z)^\top B(s', a') = B(s', a')^\top F(s, a, z)$. Therefore, the right-hand-side above rewrites as

$$\int B(s', a')B(s', a')^\top F(s, a, z)\rho(\mathrm{d}s', \mathrm{d}a') = (\mathrm{Cov}\,B)F(s, a, z) \qquad (150)$$

so that

$$(\mathrm{Cov}\,B)F(s, a, z) = \int B(s', a')m(s, a, s', a')\rho(\mathrm{d}s', \mathrm{d}a'). \qquad (151)$$

Unfolding the definition of $m$ yields the statement for $F$. The proof for $B$ is similar. $\qquad \square$

*Proof of Theorem 14.* According to the proof of Theorem 13, if $F$ achieves a local extremum of $\ell(F, B)$ given a fixed $B$, we have

$$(\mathrm{Cov}\,B)F(s, a, z) = \int B(s', a')m^{\pi_z}(s, a, s', a')\rho(\mathrm{d}s', \mathrm{d}a') \qquad (152)$$

where $m^{\pi_z}(s, a, s', a') = \sum_{t \geq 0} \gamma^t \frac{P_t(\mathrm{d}s', \mathrm{d}a'|s, a, \pi_z)}{\rho(\mathrm{d}s', \mathrm{d}a')}$ is the successor state density induced by the policy $\pi_z$.

Let $r \colon S \times A \to \mathbb{R}$ a bounded reward function that lies in the span of $B$ i.e there exists $\omega \in \mathbb{R}^d$ such that $r(s, a) = B(s, a)^\top \omega$ for any state-action pair $(s, a)$. This implies that $(\mathrm{Cov}\,B)\omega = \mathbb{E}_{(s,a)\sim\rho}[B(s, a)B(s, a)^\top \omega] = \mathbb{E}_{(s,a)\sim\rho}[B(s, a)r(s, a)] = z_R$, by definition of $z_R$.

Therefore,

$$F(s, a, z_R)^\top z_R = F(s, a, z_R)^\top (\mathrm{Cov}\,B)\omega \qquad (153)$$

$$= ((\mathrm{Cov}\,B)F(s, a, z_R))^\top \omega \qquad (154)$$

$$= \left( \int B(s', a')^\top m^{\pi_{z_R}}(s, a, s', a')\rho(\mathrm{d}s', \mathrm{d}a') \right) \omega \qquad (155)$$

$$= \int B(s', a')^\top \omega\, m^{\pi_{z_R}}(s, a, s', a')\rho(\mathrm{d}s', \mathrm{d}a') \qquad (156)$$

$$= \int r(s', a')m^{\pi_{z_R}}(s, a, s', a')\rho(\mathrm{d}s', \mathrm{d}a') \qquad (157)$$

$$= Q^{\pi_{z_R}}(s, a) \qquad (158)$$

Therefore, $\pi_{z_R}$ is the greedy policy with respect to its own Q-value. We conclude that $\pi_{z_R}$ is optimal for $r$.

Now, let $r \colon S \times A \to \mathbb{R}$ be an arbitrary bounded reward function, and let $r_B$ be the $L^2(\rho)$-projection of $r$ onto the span of $B$. Both $r$ and $r_B$ share the same $z_R = \mathbb{E}_{(s,a) \sim \rho}[r(s,a)B(s,a)] = \mathbb{E}_{(s,a) \sim \rho}[r_B(s,a)B(s,a)]$. According to the first part of our proof, $\pi_{z_R}$ is optimal for $r_B$ since $r_B$ lies in the span of $B$. Denote by the subscript $r$ in $V_r$ the reward function that a value function corresponds to. By Proposition 19,

$$0 \leq V_r^{\star} - V_r^{\pi_{z_R}} \leq (\mathrm{Id} - \gamma P_{\pi^{\star}})^{-1}(r - r_B) - (\mathrm{Id} - \gamma P_{\pi_{z_R}})^{-1}(r - r_B) \tag{159}$$

hence, taking norms,

$$\|V^{\pi_{z_R}} - V^{\star}\|_{\infty} \leq \left\|(\mathrm{Id} - \gamma P_{\pi^{\star}})^{-1}(r - r_B)\right\|_{\infty} + \left\|(\mathrm{Id} - \gamma P_{\pi_{z_R}})^{-1}(r - r_B)\right\|_{\infty} \tag{160}$$

as needed.

The bound with $\frac{2}{1-\gamma}$ follows by noting that $(\mathrm{Id} - \gamma P_{\pi})^{-1}$ is bounded by $\frac{1}{1-\gamma}$ in $L^{\infty}$ norm for any policy $\pi$. $\qquad\square$

*Proof of Proposition 15.* From the definition (42) of $\hat{z}_R$ and the definition (43) of $\hat{r}$, we find

$$\hat{z}_R = \mathbb{E}_{\rho}[B(s,a)\hat{r}(s,a)] \tag{161}$$
$$= \mathbb{E}_{\rho}[B(s,a)B(s,a)^{\top}w] \tag{162}$$
$$= (\mathrm{Cov}_{\rho} B)w \tag{163}$$

hence the result given the expression (43) for $w$.

If $\rho = \rho_{\text{test}}$, then the covariances cancel out: we find $\hat{z}_R = (\mathrm{Cov}_{\rho} B)w = (\mathrm{Cov}_{\rho} B)(\mathrm{Cov}_{\rho_{\text{test}}} B)^{-1} \mathbb{E}_{\rho_{\text{test}}}[r(s,a)B(s,a)] = \mathbb{E}_{\rho_{\text{test}}}[r(s,a)B(s,a)] = \mathbb{E}_{\rho}[r(s,a)B(s,a)] = z_R$.

If $r$ is linear in $B$, then the linear regression model does not depend on the data distribution $\rho_{\text{test}}$ used: if $r(s,a) = B(s,a)^{\top}w_0$ then $w = w_0$ for any $\rho_{\text{test}}$, as long as $\mathrm{Cov}_{\rho_{\text{test}}} B$ is invertible. In that case, both $z_R$ and $\hat{z}_R$ are equal to $(\mathrm{Cov}_{\rho} B)w_0$. $\qquad\square$

# D   Experimental Setup

In this section we provide additional information about our experiments.

## D.1   Environments

- **Discrete maze:** is the $11 \times 11$ classical tabular gridworld with foor rooms. States are represented by one-hot unit vectors, $S = \{0, 1\}^{121}$. There are five available actions , $A = \{\texttt{left}, \texttt{right}, \texttt{up}, \texttt{down}, \texttt{do nothing}\}$. The dynamics are deterministic and the walls are impassable.

- **Continuous maze:** is a two dimensional environment with impassable walls. States are represented by their Cartesian coordinates $(x, y) \in S = [0, 1]^2$. There are five available actions, $A = \{\texttt{left}, \texttt{right}, \texttt{up}, \texttt{down}, \texttt{do nothing}\}$. The execution of one of the actions moves the agent $0.1$ units in the desired direction, and normal random noise with zero mean and standard deviation $0.01$ is added to the position of the agent (that is, a move along the x axis would be $x' = x \pm 0.1 + \mathcal{N}(0, 0.01)$, where $\mathcal{N}(0, 0.01)$ is a normal variable with mean 0 and standard deviation $0.01$). If after a move the agent ends up outside of $[0, 1]^2$, the agent's position is clipped (e.g if $x < 0$ then we set $x = 0$). If a move make the agent cross an interior wall, this move is undone. For all algorithms, we convert a state $s = (x, y)$ into feature vector $\varphi(s) \in \mathbb{R}^{441}$ by computing the activations of a regular $21 \times 21$ grid of radial basis functions at the point $(s, y)$. Especially, we use Gaussian functions: $\varphi(s) = \left( \exp(-\frac{(x-x_i)^2+(y-y_i)^2}{\sigma}), \ldots, \exp(-\frac{(x-x_{441})^2+(y-y_{441})^2}{2\sigma^2}) \right)$ where $(x_i, y_i)$ is the center of the $i^{th}$ Gaussian and $\sigma = 0.05$.

- **FeatchReach:** is a variant of the simulated robotic arm environment from [PAR+18] using discrete actions instead of continuous actions. States are 10-dimensional vectors consisting of positions and velocities of robot joints. We discretise the original 3-dimensional action space into 6 possible actions using action stepsize of 1 (The same way as in https://github.com/paulorauber/hpg, the implementation of hindsight policy gradient [RUMS18]). The goal space is 3-dimensional space representing of the position of the object to reach.

- **Ms. Pacman:** is a variant of the Atari 2600 game Ms. Pacman [BNVB13], where an episode ends when the agent is captured by a monster [RUMS18]. States are obtained by processing the raw visual input directly from the screen. Frames are preprocessed by cropping, conversion to grayscale and downsampling to $84 \times 84$ pixels. A state $s_t$ is the concatenation of $(x_{t-12}, x_{t-8}, x_{t-4}, x_t)$ frames, i.e. an $84 \times 84 \times 4$ tensor. An action repeat of 12 is used. As Ms. Pacman is not originally a multi-goal domain, we define the set of goals as the set of the 148 reachable coordinate pairs $(x, y)$ on the screen; these can be reached only by learning to avoid monsters. In contrast with [RUMS18], who use a heuristic to find the agent's position from the screen's pixels, we use the Atari annotated RAM interface wrapper [ARO+19].

## D.2   Architectures

We use the same architecture for discrete maze, continuous maze and FeatchReach. Both forward and backward networks are represented by a feedforward neural network with three hidden layers, each with 256 ReLU units. The forward network receives a concatenation of a state and a $z$ vector as input and has $|A| \times d$ as output dimension. The backward network receives a state as input (or gripper's position for FeatchReach) and has $d$ as output dimension. For goal-oriented DQN, the $Q$-value network is also a feedforward neural network with three hidden layers, each with 256 ReLU units. It receives a concatenation of a state and a goal as input and has $|A|$ as output dimension.

For Ms. Pacman, the forward network is represented by a convolutional neural network given by a convolutional layer with 32 filters ($8 \times 8$, stride 4); convolutional layer with 64 filters ($4 \times 4$, stride 2); convolutional layer with 64 filters ($3 \times 3$, stride 1); and three fully-connected layers, each with 256 units. We use ReLU as activation function. The $z$ vector is concatenated with the output of the third convolutional layer. The output dimension of the final linear layer is $|A| \times d$. The backward network acts only on agent's position, a 2-dimensional input. It is represented by a feedforward neural network with three hidden layers, each with 256 ReLU units. The output dimension is $d$. For

goal-oriented DQN, the $Q$-value network is represented by a convolutional neural network with the same architecture as the one of the forward network. The goal's position is concatenated with the output of the third convolutional layer. The output dimension of the final linear layer is $|A|$.

### D.3 Implementation Details

For all environments, we run the algorithms for 800 epochs. Each epoch consists of 25 cycles where we interleave between gathering some amount of transitions, to add to the replay buffer $\mathcal{D}$ (old transitions are thrown when we reach the maximum of its size), and performing 40 steps of stochastic gradient descent on the model parameters. To collect transitions, we generate episodes using some behavior policy. For both mazes, we use a uniform policy while for FetchReach and Ms. Pacman, we use an $\varepsilon$-greedy policy ($\varepsilon = 0.2$) with respect to the current approximation $F(s, a, z)^\top z$ for a sampled $z$. At evaluation time, $\varepsilon$-greedy policies are also used, with a smaller $\varepsilon = 0.02$ for all environments except from discrete maze where we use Boltzmann policy with temperature $\tau = 1$. We train each model for three different random seeds.

For generality, we will keep using the notation $B(s, a)$ while in our experiments $B$ acts only on $\varphi(s, a)$, a part of the state-action space. For discrete and continuous mazes, $\varphi(s, a) = s$, for FetchReach, $\varphi(s, a)$ the position of arm's gripper and for Ms. Pacman, $\varphi(s, a)$ is the 2-dimensional position $(x, y)$ of the agent on the screen.

We denote by $\theta$ and $\omega$ the parameters of forward and backward networks respectively and $\theta^-$ and $\omega^-$ the parameters of their corresponding target networks. Both $\theta^-$ and $\omega^-$ are updated after each cycle using Polyak averaging; i.e $\theta^- \leftarrow \alpha \theta^- + (1 - \alpha)\theta$ and $\omega^- \leftarrow \alpha \omega^- + (1 - \alpha)\omega$ where $\alpha = 0.95$ is the Polyak coefficient.

During training, we sample $z$ from a rescaled Gaussian that we denote $\nu$. With a pure Gaussian in large dimension, the norm of $z$ would be very concentrated around a single value. Instead, we first sample a $d$-dimensional standard Gaussian variable $x \sim \mathcal{N}(0, \mathrm{Id}) \in \mathbb{R}^d$ and a scalar centered Cauchy variable $u \in \mathbb{R}$ of scale 0.5, then we set $z = \sqrt{d}\, u\, \frac{x}{\|x\|}$. We use a Cauchy distribution to ensure that the norm of $z$ spans the non-negative real numbers space while having a heavy tail. We also scale by $\sqrt{d}$ to ensure that each component of $z$ has an order of magnitude of 1.

Before being fed to $F$, $z$ is preprocessed by $z \leftarrow \frac{z}{\sqrt{1+\|z\|_2^2/d}}$; this way, $z$ ranges over a bounded set in $\mathbb{R}^d$, and this takes advantage of optimal policies being equal for a reward $R$ and for $\lambda R$ with $\lambda > 0$.

To update network parameters, we compute an empirical loss by sampling 3 mini-batches, each of size $b = 128$, of transitions $\{(s_i, a_i, s_{i+1})\}_{i \in I} \subset \mathcal{D}$, of target state-action pairs $\{(s_i', a_i')\}_{i \in I} \subset \mathcal{D}$ and of $\{z_i\}_{i \in I} \sim \nu$:

$$
\mathcal{L}(\theta, \omega) = \frac{1}{2b^2} \sum_{i,j \in I^2} \left( F_\theta(s_i, a_i, z_i)^\top B_\omega(s_j', a_j') - \gamma \sum_{a \in A} \pi_{z_i}(a \mid s_{i+1}) \cdot F_{\theta^-}(s_{i+1}, a, z_i)^\top B_{\omega^-}(s_j', a_j') \right)^2
$$
$$
- \frac{1}{b} \sum_{i \in I} F_\theta(s_i, a_i, z_i)^\top B_\omega(s_i, a_i) \tag{164}
$$

where we use the Boltzmann policy $\pi_{z_i}(\cdot \mid s_{i+1}) = \mathtt{softmax}(F_{\theta^-}(s_{i+1}, \cdot, z_i)^\top z_i / \tau)$ with fixed temperature $\tau = 200$ to avoid the instability and discontinuity caused by the argmax operator.

Since there is unidentifiability between $F$ and $B$ (Appendix, Remark 7), we include a gradient to make $B$ closer to orthonormal, $\mathbb{E}_{(s,a)\sim\rho} B(s, a)B(s, a)^\top \approx \mathrm{Id}$:

$$
\frac{1}{4} \partial_\omega \left\| \mathbb{E}_{(s,a)\sim\rho} B_\omega(s, a)B_\omega(s, a)^\top - \mathrm{Id} \right\|^2 =
$$
$$
\mathbb{E}_{(s,a)\sim\rho,(s',a')\sim\rho}\, \partial_\omega B_\omega(s, a)^\top \left( B_\omega(s, a)^\top B_\omega(s', a') \cdot B_\omega(s', a') - B(s, a) \right) \tag{165}
$$

To compute an unbiased estimate of the latter gradient, we use the following auxiliary empirical loss:

$$\mathcal{L}_{\texttt{reg}}(\omega) = \frac{1}{b^2} \sum_{i,j \in I^2} B_\omega(s_i, a_i)^\top \texttt{stop-gradient}(B_\omega(s'_j, a'_j)) \cdot \texttt{stop-gradient}(B_\omega(s_i, a_i)^\top B_\omega(s'_j, a'_j))$$

$$- \frac{1}{b} \sum_{i \in I} B_\omega(s_i, a_i)^\top \texttt{stop-gradient}(B_\omega(s_i, a_i)) \tag{166}$$

Finally, we use the Adam optimizer and we update $\theta$ and $\omega$ by taking a gradient step on $\mathcal{L}(\theta, \omega)$ and $\mathcal{L}(\theta, \omega) + \lambda \cdot \mathcal{L}_{\texttt{reg}}(\omega)$ respectively, where $\lambda$ is a regularization coefficient that we set to 1 for all experiments.

We summarize the hyperparameters used for FB algorithm and goal-oriented DQN in table 1 and 2 respectively.

| Hyperparameters | Discrete Maze | Continuous Maze | FetchReach | Ms. Pacman |
|---|---|---|---|---|
| number of cycles per epoch | 25 | 25 | 25 | 25 |
| number of episodes per cycles | 4 | 4 | 2 | 2 |
| number of timesteps per episode | 50 | 30 | 50 | 50 |
| number of updates per cycle | 40 | 40 | 40 | 40 |
| exploration $\varepsilon$ | 1 | 1 | 0.2 | 0.2 |
| evaluation $\varepsilon$ | Boltzman with $\tau = 1$ | 0.02 | 0.02 | 0.02 |
| temperature $\tau$ | 200 | 200 | 200 | 200 |
| learning rate | 0.001 | 0.0005 | 0.0005 | 0.0001 if $d = 100$ else 0.0005 |
| mini-batch size | 128 | 128 | 128 | 128 |
| regularization coefficient $\lambda$ | 1 | 1 | 1 | 1 |
| Polyak coefficient $\alpha$ | 0.95 | 0.95 | 0.95 | 0.95 |
| discount factor $\gamma$ | 0.99 | 0.99 | 0.9 | 0.9 |
| replay buffer size | $10^6$ | $10^6$ | $10^6$ | $10^6$ |

Table 1: Hyperparameters of the FB algorithm

| Hyperparameters | Discrete Maze | Continuous Maze | FetchReach | Ms. Pacman |
|---|---|---|---|---|
| number of cycles per epoch | 25 | 25 | 25 | 25 |
| number of episodes per cycles | 4 | 4 | 2 | 2 |
| number of timesteps per episode | 50 | 30 | 50 | 50 |
| number of updates per cycle | 40 | 40 | 40 | 40 |
| exploration $\varepsilon$ | 0.2 | 0.2 | 0.2 | 0.2 |
| evaluation $\varepsilon$ | Boltzman with $\tau = 1$ | 0.02 | 0.02 | 0.02 |
| learning rate | 0.001 | 0.0005 | 0.0005 | 0.0005 |
| mini-batch size | 128 | 128 | 128 | 128 |
| Polyak coefficient $\alpha$ | 0.95 | 0.95 | 0.95 | 0.95 |
| discount factor $\gamma$ | 0.99 | 0.99 | 0.9 | 0.9 |
| replay buffer size | $10^6$ | $10^6$ | $10^6$ | $10^6$ |
| ratio of hindsight replay | - | - | - | 0.8 |

Table 2: Hyperparameters of the goal-oriented DQN algorithm

## D.4 Experimental results

In this section, we provide additional experimental results.

### D.4.1 Goal-Oriented Setup: Quantitative Comparisons

### D.4.2 More Complex Rewards: Qualitative Results

### D.4.3 Embedding Visualization

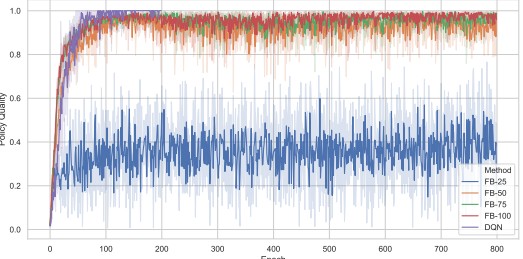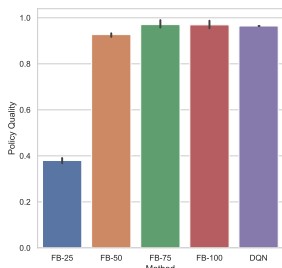

Figure 8: **Discrete maze**: Comparative performance of FB for different dimensions and DQN. **Left**: the policy quality averaged over 20 randomly selected goals as function of the training epochs. **Right**: the policy quality averaged over the goal space after 800 training epochs.

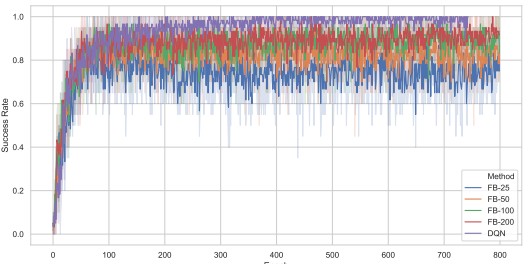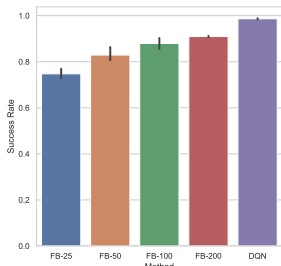

Figure 9: **Continuous maze:** Comparative performance of FB for different dimensions and DQN. **Left**: the success rate averaged over 20 randomly selected goals as function of the training epochs. **Right**: the success rate averaged over 1000 randomly sampled goals after 800 training epochs.

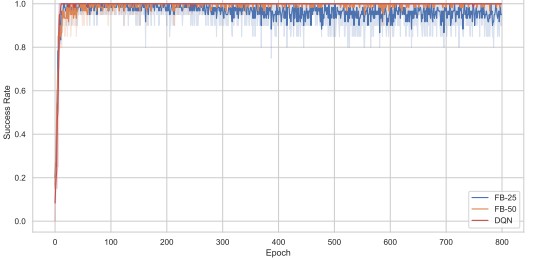

Figure 10: **FetchReach**: Comparative performance of FB for different dimensions and DQN. **Left**: the success rate averaged over 20 randomly selected goals as function of the training epochs. **Right**: the success rate averaged over 1000 randomly sampled goals after 800 training epochs.

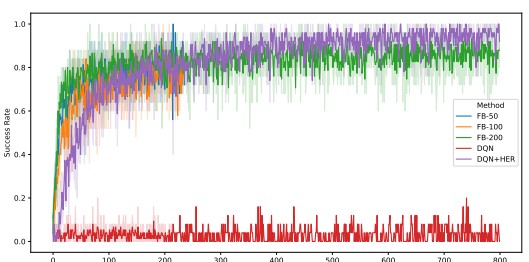 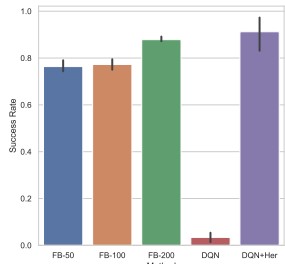

Figure 11: **Ms. Pacman**: Comparative performance of FB for different dimensions and DQN. **Left**: the success rate averaged over 20 randomly selected goals as function of the training epochs. **Right**: the success rate averaged over the 184 handcrafted goals after training epochs. Note that FB-50 and F-100 have been trained only for 200 epochs.

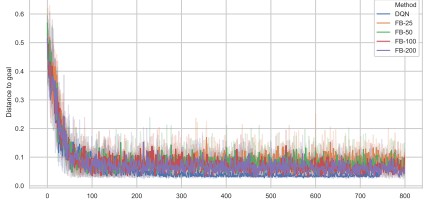 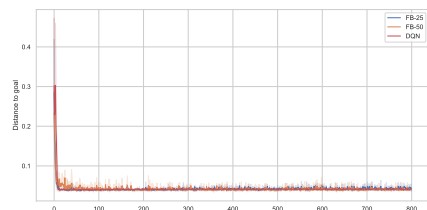

Figure 12: Distance to goal of FB for different dimensions and DQN as function of training epochs. **Left**: Continuous maze. **Right**: FetchReach.

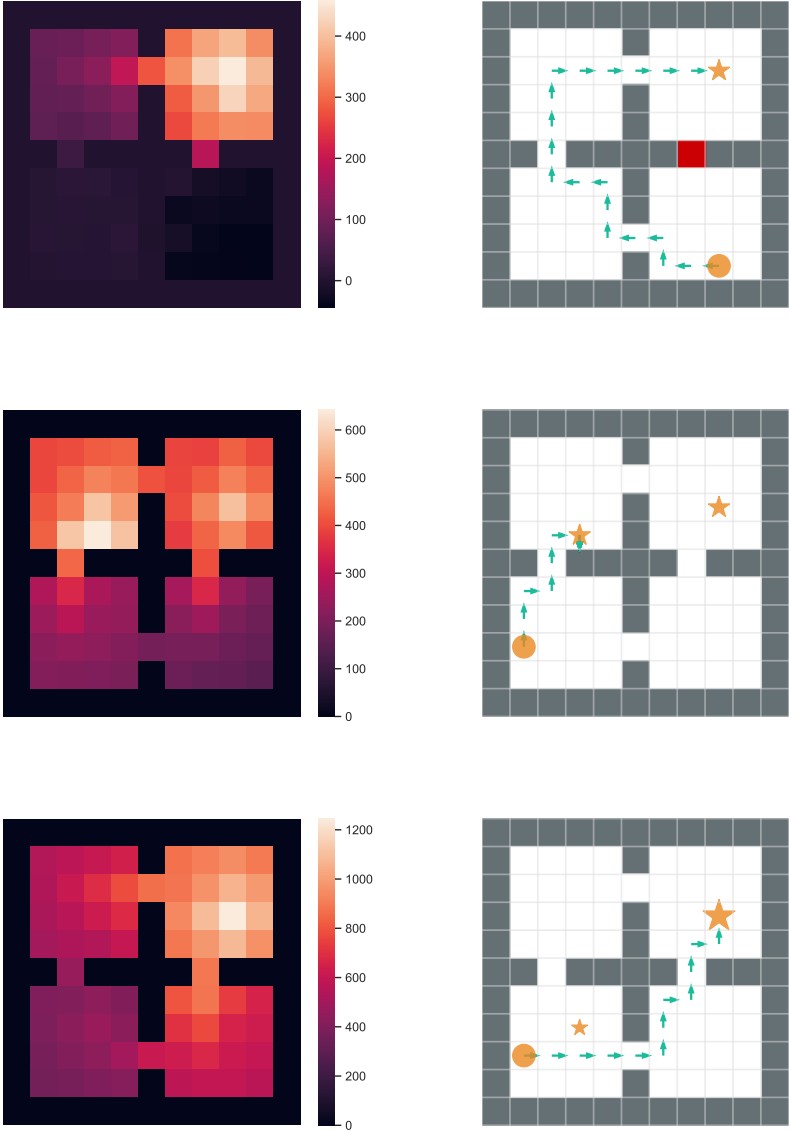

Figure 13: **Discrete Maze**: Heatmap plots of $\max_{a \in A} F(s, a, z_R)^\top z_R$ (left) and trajectories of the Boltzmann policy with respect to $F(s, a, z_R)^\top z_R$ with temperature $\tau = 1$ (right). **Top row**: for the task of reaching a target while avoiding a forbidden region, **Middle row**: for the task of reaching the closest goal among two equally rewarding positions, **Bottom row**: choosing between a small, close reward and a large, distant one.

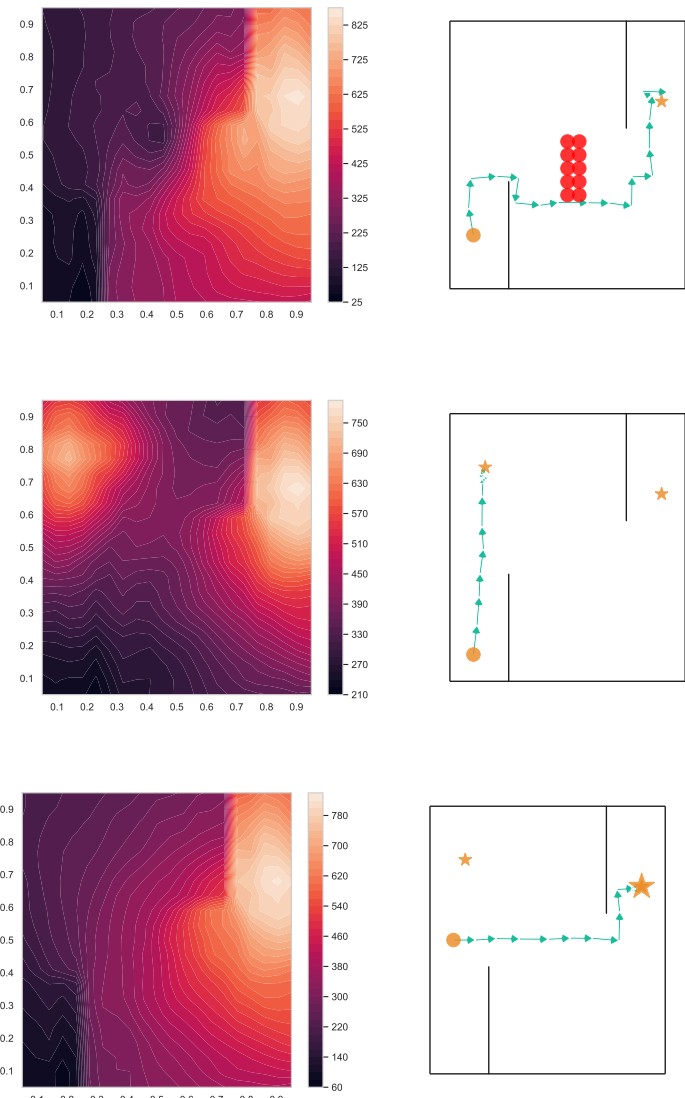

Figure 14: **Continuous Maze**: Contour plots plot of $\max_{a \in A} F(s, a, z_R)^\top z_R$ (left) and trajectories of the $\varepsilon$ greedy policy with respect to $F(s, a, z_R)^\top z_R$ with $\varepsilon = 0.1$ (right). **Left**: for the task of reaching a target while avoiding a forbidden region, **Middle**: for the task of reaching the closest goal among two equally rewarding positions, **Right**: choosing between a small, close reward and a large, distant one..

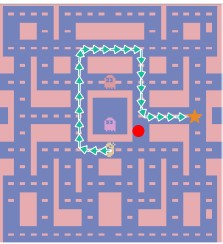 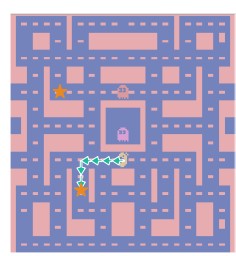 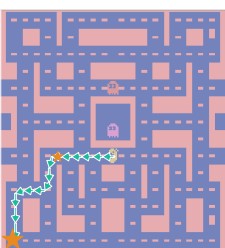

Figure 15: **Ms. Pacman**: Trajectories of the $\varepsilon$ greedy policy with respect to $F(s, a, z_R)^\top z_R$ with $\varepsilon = 0.1$ (right). **Top row**: for the task of reaching a target while avoiding a forbidden region, **Middle row**: for the task of reaching the closest goal among two equally rewarding positions, **Bottom row**: choosing between a small, close reward and a large, distant one..

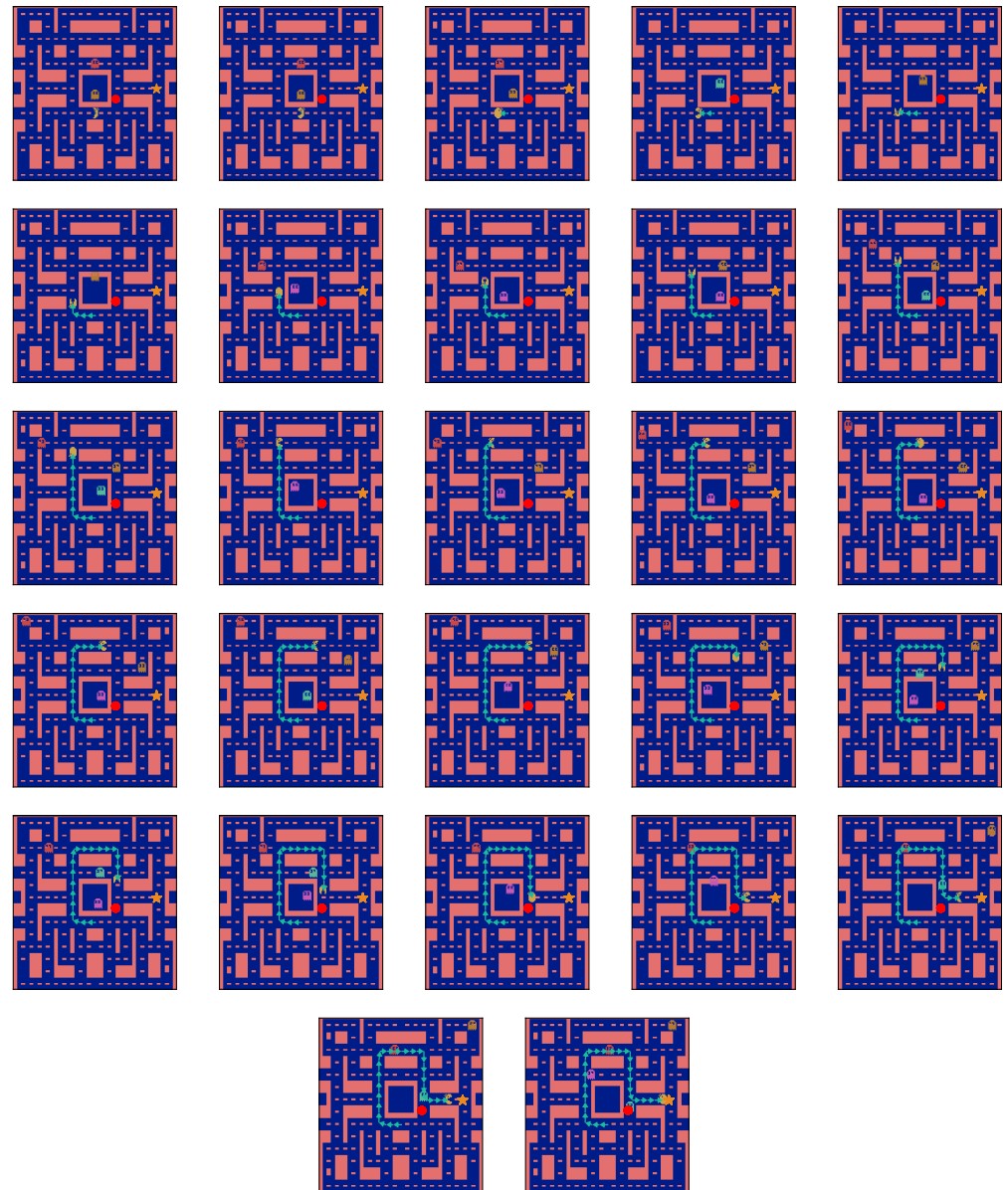

Figure 16: Full series of frames in Ms. Pacman along the trajectory generated by the $F^\top B$ policy for the task of reaching a target position (star shape ⭐) while avoiding forbidden positions (red shape 🔴).

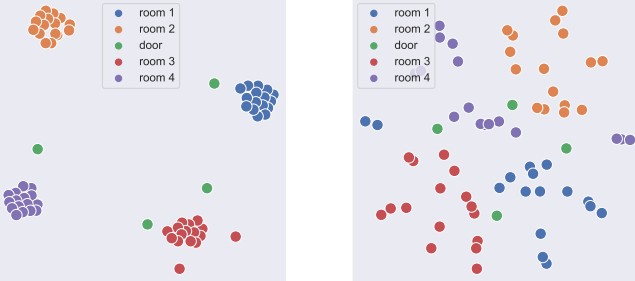

Figure 17: **Discrete maze**: Visualization of FB embedding vectors after projecting them in two-dimensional space with t-SNE. **Left**: the $F$ embedding for $z = 0$. **Right**: the $B$ embedding. Note how both embeddings recover the foor-room and door structure of the original environment. The spread of B embedding is due to the regularization that makes B closer to orthonormal.

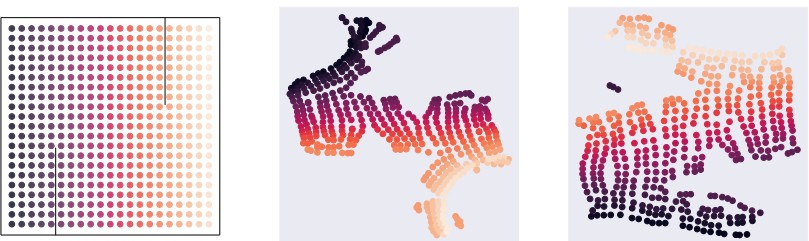

Figure 18: **Continuous maze**: Visualization of FB embedding vectors after projecting them in two-dimensional space with t-SNE. **Left**: the states to be mapped. **Middle**: the $F$ embedding. **Right**: the $B$ embedding.

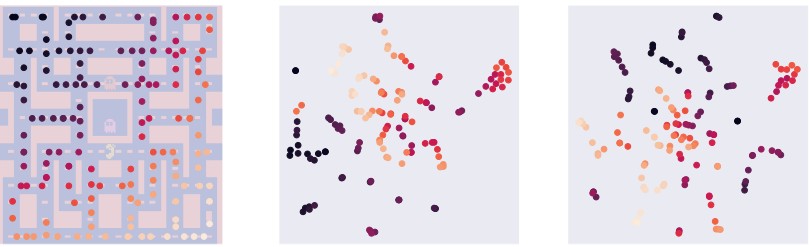

Figure 19: **Ms. Pacman**: Visualization of FB embedding vectors after projecting them in two-dimensional space with t-SNE. **Left**: the agent's position corresponding to the state to be mapped. **Middle**: the $F$ embedding for $z = 0$. **Right**: the $B$ embedding. Note how both embeddings recover the cycle structure of the environment. F acts on visual inputs and B acts on the agent's position.

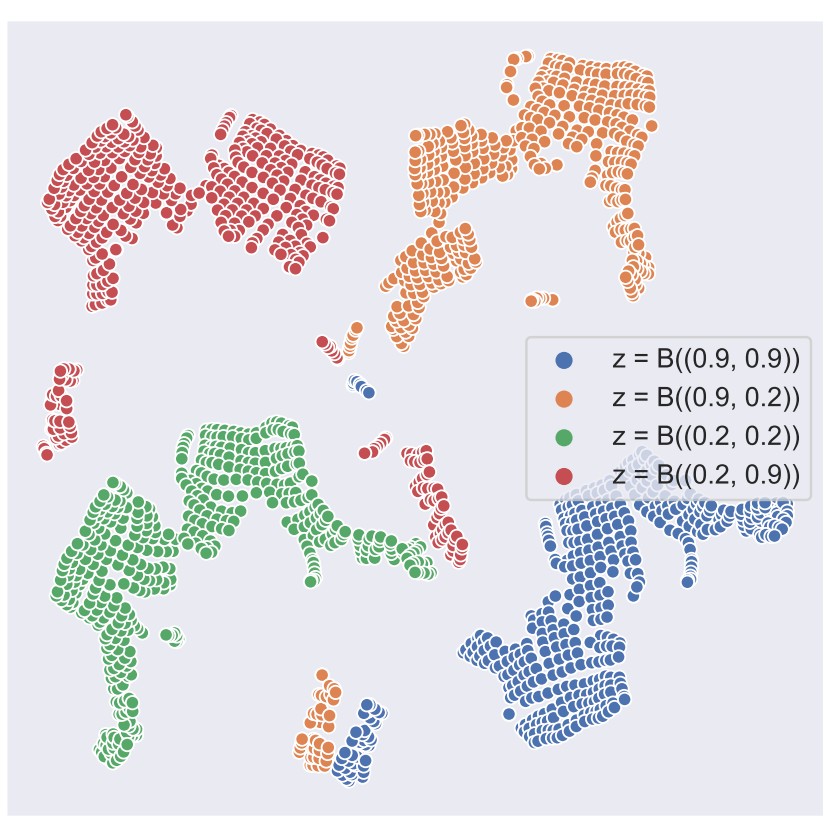

Figure 20: **Continuous maze**: visualization of $F$ embedding vectors for different $z$ vectors, after projecting them in two-dimensional space with t-SNE.