# OpenReview forum: "Learning One Representation to Optimize All Rewards"
_NeurIPS.cc/2021/Conference — NeurIPS 2021 Poster_

### Official Review · Reviewer_Bh3C · 2021-07-13

**Rating:** 5
**Confidence:** 5

**Summary:**

This paper proposes a new representation in MDPs such that it can be learned using transition samples and used to compute the optimal policy. I am not convinced by the motivation of such a new representation since no computational or statistical efficiency results are established and thus suggest not accepting the paper.

**Limitations And Societal Impact:**

yes

**Main Review:**

This paper proposes a new representation in MDPs such that it can be learned using transition samples and used to compute the optimal policy.

My main concern is that such a representation might be too restrictive for learning unknown MDPs. As one can always learn the transition kernel well without knowing any reward signals and later compute an optimal policy by a dynamic program procedure using the learned model and reward function provided a posterior, the proposed new algorithm just provides an alternative which might save some computation in the second step while not sure adds how much restriction on the first step. For the tabular MDPs, the dimension d of is already |SA| as shown in the appendix and I am not sure how large the computation cost can be to learn the map F: S $\times$ A $\times$ Z $\rightarrow$ Z and B. To me, it feels equal to or more expensive than learning the transition kernel matrix. Also, it is not clear whether this new representation can help improve the exploration efficiency in the reward-free exploration phase, which is usually the result of many other reward-free works, e.g., see [1].

In all, I am not convinced by the motivation of such a new representation since no computational or statistical efficiency results are established and thus suggest not accepting the paper.

[1] Jin, Chi, Akshay Krishnamurthy, Max Simchowitz, and Tiancheng Yu. "Reward-free exploration for reinforcement learning." In International Conference on Machine Learning, pp. 4870-4879. PMLR, 2020.

**Time Spent Reviewing:**

3

---

> ### Author Response · Authors · 2021-08-06
> **Response to reviewer Bh3C**
>
> Thank you for your comments and the time spent on our paper.
>
> We believe we consider a quite different (and complementary) problem wrt the reference [1].
>
> In our setup, "learn the transition kernel, then solve the planning problem for each new target/reward" is not a feasible solution.  Indeed we are interested in reducing (to almost 0) the computation time of the planning step each time a new target/reward is given. We are not interested in computing "the" optimal policy, but many optimal policies on the fly, or even an infinite number of optimal policies (e.g., a robot learning to reach an arbitrary state when instructed to do so requires representing all optimal policies for every possible target state). We apologize if this was unclear in the text.
>
> For instance, in robotics, a model of the one-timestep transition function is often easy to learn and may even be known explicitly. But it is computationally too intense to solve a new planning problem every time a new target state is given. The passage from knowing a one-step transition function to knowing the target-dependent Q-function is highly nontrivial in practice.
>
> This is why we are interested in building an unsupervised representation of the MDP, on which (approximate) solutions to all planning problems can be read directly without additional computation. We also want this representation to work with function approximation, contrary to [1] which is specific to finite spaces.
>
> Thank you for asking why learning an FB model would work better than learning a model of the transition matrix P in practice. First, a model of P has to be generative: we have to synthesize the next state, and generative models are difficult to train for complex objects. Second, planning based on P relies on iterating the one-step model, and any approximation errors in P get compounded quickly when generating synthetic sequences of states.
>
> The FB model is a low-rank approximation of the resolvent matrix $\sum \gamma^t P^t=(Id-\gamma P)^{-1}$ [for a fixed policy; the additional z in F makes the policy variable]. Our algorithm for learning this resolvent matrix does not need generative modeling and unrolling synthetic trajectories.
>
> This low-rank approximation is also relevant to your question about the dimension d. As you say, an exact solution requires $d=S \times A$ in a finite space. But this is only needed for an exact (full-rank) representation of the resolvent matrix of the MDP. A small rank d still provides useful behaviors.  For instance, we conducted experiments for navigation on a k-dimensional discrete grid $[1 \dots 10]^k$ of side 10, and we found out that taking d of order 2k or 3k is enough to learn to reach any goal, even though the number of states is $10^k$ and the true rank is huge. We can include these experiments in the updated version.
>
> Fundamentally, this works because a low-rank approximation makes much more sense for the resolvent than for $P$ itself. In the resolvent, large eigenvalues/eigenvectors of $P$ stand out ($P^t$ is dominated by the main eigenvectors), while P itself generally has no useful low-rank approximations. This is already clear, eg., letting $P$ be the random walk on a cyclic graph [1...n]. The first eigenvectors of the resolvent already contain a lot of useful information on the MDP, as in the example of the cube above.
>
> Thus, learning a model of the resolvent is not particularly more restrictive than learning a model of $P$ for unknown MDPs. It has the advantage of directly providing the optimal policies (thanks to the additional dependency on $z$ in $F$) without planning.
>
> True, contrary to [1] we do not deal with sample complexity bounds for exploration. But to our knowledge, such bounds are rarely available for deep reinforcement learning methods with function approximation. Finally, our method is complementary to [1] in that [1] builds a dataset of transitions with good statistical properties in a finite MDP, but relies on an external planner for computing policies; meanwhile, our method assumes access to a good existing dataset of transitions or exploration policy, and, based on this, builds a summary from which any optimal policy for any reward can be obtained immediately without any planning.

---

### Official Review · Reviewer_MgL9 · 2021-07-16

**Rating:** 8
**Confidence:** 2

**Summary:**

This paper proposes an algorithm that is able to learn the inherent structure of a reward-free MDP in a compact form, which can quickly give a nearly optimal policy after the reward function is specified. The algorithm is developed through a rigorous theoretical analysis and the experiment also shows comparable or even better performance than the algorithm trained with specified rewards.

**Limitations And Societal Impact:**

The authors adequately addressed the limitations of their work in Section 4 by pointing out the potential problems of their algorithm. The potential negative social impact is not addressed, which will generally be all potential malicious usage of any reinforcement learning algorithms.

**Main Review:**

Overall, I appreciated the novel and rigorous theoretical framework developed for learning MDP representation in this paper. Meanwhile, the experimental result looks promising and the extended analysis in appendix looks thorough. However, the writing of this paper may need some improvement since currently the ordering of some content makes it harder to understand, details in "Suggestion on writing" part.

Meanwhile, I also have the following concerns and questions.

Questions:
- On page 6, why it is okay to write $z_R$ as stated in Eq. (9) when the reward is known explicitly? Is Eq. (9) consistent with the definition of $z_R$ in Eq. (8)?
- In the experiment for FetchReach and Ms. Pacman, the behavior policy becomes adaptive, which implies a changing distribution $\rho$; meanwhile, the theory uses a fixed distribution $\rho$. Is there any intuition on why a changing distribution $\rho$ is better under this scenario?
- Does the distribution of $z$ play any role in the theory development? It seems $z$ is always fixed in theory while the distribution $\nu$ of $z$ is very carefully designed in experiment.

Suggestions on writing:
- At page 5, it may be better to explicitly point out that Theorem 6 in [BTO21] gives the update rule in Eq. (7).
- It may be better to move the explanations about $\mathscr{L}$ and $\mathscr{L}_{\mathsf{reg}}$ at Section D.3 to Section A right below the full Algorithm 1. Or more aggressively, it may be better to move all of these into the main text of Section 4, which can help a lot more in understanding the algorithm than reading the paragraph description in current Section 4. This extra space can be saved from moving some figures and experimental results into the Appendix.
- In Remark 4, page 17, it may be better to say explicitly that $B'$ being $L^2(\rho)$-orthonormal means $\mathbb{E}_{(s, a)\sim\rho}\left[B'(s, a)B'(s, a)^\top\right]=\mathbf{I}$ if I understand correctly.

```
[BTO21] Léonard Blier, Corentin Tallec, and Yann Ollivier. Learning successor states and goal-dependent values: A mathematical viewpoint. arXiv preprint arXiv:2101.07123, 2021.
```

---

Good paper. My score remains the same after rebuttal.

**Time Spent Reviewing:**

7

---

> ### Author Response · Authors · 2021-08-06
> **Response to reviewer MgL9**
>
> Thank you for your detailed comments and questions, and for the time spent reviewing.
>
> About the distribution of $z$: ideally, the equality $FB=m$ should be valid for all values of $z$ (the loss should be 0 for every $z$). For practical learning, we have to sample $z$ from some distribution, and we chose a distribution that covers the $Z$ space well. We do not have a rigorous theory for that, but we have strong hints: At test time, a typical goal-oriented application will use the policies $\pi_z$ with $z=B(s)$ for a variety of target states $s$.  So it makes sense to match the training distribution of $z$ with the distribution of the values $B(s)$ for random states $s$. Thus, during training, it makes sense to use $Cov B(s)$ as the covariance of $z$. During training, we normalize $Cov B(s)$ to $Id$ for purely numerical reasons (mathematically this is neutral by Remark 4). Hence a unit covariance for $z$: this is optimal for a goal-oriented application with target states sampled from rho.
>
> The reason our training distribution of $z$ is "very carefully designed" (a rescaled Gaussian) in the experiments is just a quirk of large-dimensional Gaussians: in large dimension d, a random Gaussian vector always has a norm very close to $\sqrt{d}$. An extra scaling factor ensures that learning occurs for a wider set of values of $|z|$.
>
> On the expression for $z_R$ in (9) vs (8): Let $\delta_x$ be the Dirac distribution defined wrt the reference measure $\rho$ (namely, by definition, the integral $\int f(y) \delta_x(y) \rho(dy) = f(x)$ for any test function f). Then the expression in (9) coincides with the value $z_R$ from (8) for the distribution-valued reward $R=\delta_{s0,a0}-\lambda \sum \delta_{s_i,a_i}$. [Note: In continuous spaces, distributions are the only meaningful way to define pointwise rewards: we have to define the reward to be infinite when reaching the target because in stochastic continuous environments the probability to exactly reach the target is 0. Infinite pointwise rewards lead to well-defined Q-functions, while just defining the reward to be 1 would lead to the Q-function being 0.] This does introduce "hidden" factors $\rho(s_i,a_i)$ in the reward, which are usually unknown. In practice, we do not think this will be a problem, and this can be adjusted manually via the constant lambda. Note that in a pure goal-oriented setting the hidden factor is harmless: scaling the reward does not change the optimal policy, so using $z=B(s_0,a_0)$ has the same optimal policy as $z=B(s_0,a_0) \rho(s_0,a_0)$.
>
> About adaptivity/changing rho in the experiments: Indeed you are correct that a distribution shift for rho occurs during learning. The current policies being learned are used for exploration purposes: this has the side effect of slowly changing the distribution of states in the replay buffer. Asymptotically, since we always add to the replay buffer, we would expect the distribution of states in the buffer to reach a stationary distribution rho, and the theory would apply to this limit rho. We do not have a rigorous theory to analyze the distribution shift. This phenomenon is not specific to our method: any Q-learning method with a replay buffer updated online will have a changing distribution of states, and in principle, the resulting change of sampling distribution will affect the underlying stochastic gradient algorithm.
>
> Finally, thank you for your comments on the writing. We will implement your suggestions as well as those of the other reviewers.

---

### Official Review · Reviewer_ojoH · 2021-07-16

**Rating:** 7
**Confidence:** 4

**Summary:**

This paper considered the representation learning in reward-free MDPs to encode the optimal policy given any reward function without planning. The proposed unsupervised learning algorithm FB offered two features $F$ and $B$ that encoded the information of the transition to extract the optimal policy. They showed that as long as $F^\top B$ approximates the probability of visiting one state-action to another, then we can extract the optimal policy given any reward function directly. Motived by the nice theoretical properties of $F$ and $B$, they showed that the unsupervised learning algorithm can achieve the same level of learning compared with DQN, DQN with HER, etc. on several benchmark environments. They also incorporated some engineering tricks to make the theory-boosted algorithm more effective.

**Limitations And Societal Impact:**

The limitations have been addressed in the paper, including the lack of exploration, the large variance, etc. The lack of exploration can be a serious problem, because rewards in real-world are mostly not random. Perhaps this problem is related to some techniques in batch RL. Moreover, it would be better to show more experimental results, such as in more realistic environments, or compared with more goal-oriented baseline algorithms.

**Main Review:**

Novelty: The paper proposed a novel representation learning idea to learn the forward-backward features $F$ and $B$.  The authors constructed $F$ and $B$ according to a novel observation that $F^\top B$ should approximate the probability of visiting one state-action to another. They also proposed new unsupervised learning method to train $F$ and $B$ based on a new density measure.

Quality & Clarity: The claims are clear and well-supported to me. The proofs are clean and correct. The organization is okay, but some statements in several sections are obscure and took quite a while to understand. For example, the authors introduced their ideas in the related work section, in a fashion of story-telling. This type of writing may fail to highlight the critical points of the paper, and make the readers confusing because they are not comfortable with the notations.

Significance: The paper studied a very important problem for both theoretical RL and empirical RL: how to “summarize” an environment, so as to remove the useless information and keep the important part. The FB representation provided valuable insights on this problem. The experiments on Maze and Ms. Pacman also verified that FB is comparable to the classic value-based methods, but FB is more flexible to random goals.


**Time Spent Reviewing:**

2

---

> ### Author Response · Authors · 2021-08-06
> **Response to reviewer ojoH**
>
> Thank you for your comments, assessment, and the time spent reviewing.
>
> We agree that our method does not solve exploration, but we consider this as a separate line of research requiring other ideas. (Our method does provide additional tools for exploration, such as using the variety of policies being learned for different values of z, as we do in the experiments; still, this assumes some basic exploration has already occurred.)
>
> You are not the only reviewer to complain about some sections being difficult to understand. We will reorganize the text taking into account your remarks as well as those of the other reviewers.

---

### Official Review · Reviewer_NkUB · 2021-07-21

**Rating:** 7
**Confidence:** 4

**Summary:**

This paper proposes an algorithm learning a kind of universal value function (UVFA), so that given a buffer with a good coverage of transitions it returns the optimal behavior for any reward function or any goal state by computing a simple average.

The idea comes from the fact that the Q-function for a given policy $\pi$ -- that induces a reaching probability from any ($s,a$) $M^\pi(s, a, s', a')$ -- can be expressed as the scalar produce $Q^\pi(s,a) = \sum_{s',a'} M^\pi(s,a, s', a') r(s',a') = \langle M(s,a); r\rangle$.
Then, if we learn $F(s,a,z)$ and $B(z)$ such that $M^{\pi_z}(s,a) = F(s,a, z)^T B(z)$ we obtain $Q^\pi_z = \langle F(s,a, z)^T B(z); r\rangle = F(s,a, z_r)^T z_r$ where $z_r = \langle B(z); r\rangle$.

The greedy policy given such a Q-function should maximize (over action) $F(s,a, z_r)^T z_r$. Now, there is a fix point {$F^*, B^*, \pi^*$} such that for all $z$, $Q^{\pi^*}(s, a) = F^*(s,a, z)^T z$, which coincides with the optimal Q-function for the given $z$.

The paper provides an algorithm that learns these optimal $F^*$ and $B^*$. Then, given any reward function, they compute $z_r = \langle B^*(z); r\rangle$ and return the optimal Q-function $Q^*(s,a) = F^*(s,a, z_r)^T z_r$.


**Limitations And Societal Impact:**

This paper has no societal impact.

**Main Review:**

The idea is brilliant, and I think will have a strong impact on the topic of UVFA.

However, reading this paper was quite painful, in the sense that it is really badly constructed. A lot of quantities are introduced before the notation is explained, or ideas spoiled before the content makes it understandable. I agree that in the abstract and introduction the authors don't have the choice and must explain the idea before having the mathematical notation and tools. But --in my case-- I finally understood what the authors were talking about  at line 183 with the "sketch of proof" (page 5).
I would suggest a lot of improvements regarding the construction of the paper, but to rather summarize:

1) separate introduction and related work, and move the related work later in the paper (for ex before or after experiments)
2) improve the introduction by a better explanation "with hands" of what $F$ and $B$ represent (future and past), how to use $B$ to compute $z_r$ and then $F$ to get the optimal policy.
3) in problem and notation, give the problem after the notation.
4) right after the notation, briefly explain the idea like I did in the summary of this review.
5) put elements of your proof in the paper, so that you prepare the reader to understand the TH1
6) ends up with TH1

In the current state, we don't understand the quantities before TH1, then we don't understand TH1, then there is a long explanation that we finally understand by reading the short "sketch of proof".

Another problem of this paper regards the experiment design: did the author used more than one seed? if yes how many? (especially for the pacman expe). The pacman results are promising (FB works better than HER), it would have been great to have more examples of such complex environments.

I think this paper is already nice and important to be communicated, but I would be a bit disappointed because of its actual form (difficult to read), so I am on the fence, marginally above the acceptance.


**Time Spent Reviewing:**

6

---

> ### Author Response · Authors · 2021-08-06
> **Response to reviewer NkUB**
>
> Thank you for your detailed reading, your compliments on the idea, and your suggestions for improved writing.
>
> All the experiments were run with three different random seeds.
>
> We can (and will) certainly rework the text according to your suggestions for improved clarity. We already tried several arrangements of the definitions: however, in each case, there is a difficulty due to the circular dependency between $M$, $FB$, and $\pi$: mathematical rigor forces us to introduce several objects before they are fully explained. We will definitely follow your suggestions to expose the core algebraic idea (as explained in the "sketch of proof" and in your summary) early on, after the notation; as well as your other suggestions.
>
> Here's how we could describe the core idea after the Notation section, in addition to your other suggestions. (We will have to save space elsewhere.)
>
>
> Core idea. The main algebraic idea is as follows. For a fixed policy, the $Q$-function depends lineary on the reward: assuming for simplicity that $S$ is finite, we have $Q^\pi_r(s,a)=\sum_{s',a'} M^\pi(s,a,s',a')r(s',a')$ where $M^\pi(s,a,s',a'):= \sum_{t\geq 0} \gamma^t \Pr\left((s_t,a_t)=(s',a')|s,a,\pi\right)$.  This rewrites as $Q^\pi_r=M^{\pi}r$ viewing everything as vectors and matrices indexed by state-actions.
>
> Now let $(\pi_z)$ be any family of policies parameterized by $z \in R^d$. Assume that for each $z$, we can find $d\times (S\times A)$-matrices $F_z$ and $B$ such that $M^{\pi_z}=F_z^T B$. Then $Q_r^{\pi_z}=F_z^T Br$.  Specializing to $z_R:= Br$, the $Q$-function of policy $\pi_{z_R}$ on reward $r$ is $F_{z_R}^T z_R$. So far $\pi_z$ was unspecified; but if we define $\pi_z(s):= \arg\max_a (F_z^T z)[s,a]$ at each state $s$, then $\pi_{z_R}$ is the greedy policy with respect to $F_{z_R}^T z_R$. At the same time, $F_{z_R}^T z_R$ is the $Q$-function of $\pi_{z_R}$ for reward $r$: thus, $\pi_{z_R}$ is optimal for reward $r$.
>
> Thus, if we manage to find $F$, $B$, and $\pi_z$ such that $\pi_z=\arg\max F_z^T z$ and $F_z^T B=M^{\pi_z}$ for all $z\in R^d$, then we obtain the optimal policy for any reward $r$, just by computing $Br$ and applying policy $\pi_{Br}$.
>
> This criterion on $(F,B,\pi_z)$ is entirely unsupervised. Since $F$ and $B$ depend on $\pi_z$ but $\pi_z$ is defined via $F$, this is a fixed point equation.  We present a well-grounded algorithm to learn such $F$, $B$, and $\pi_z$.  An exact solution exists for $d$ large enough (Appendix), while a smaller $d$ provides lower-rank approximations $M^{\pi_z}\approx F_z^T B$.
>
> This extends to continuous spaces with function approximation (Thm.~1): $F_z$ and $B$ become functions $S\times A\to R^d$ instead of matrices; since $F_z$ depends on $z$, $F$ itself is a function $S\times A\times R^d\to R^d$. The sums over states are replaced with expectations under the data distribution $\rho$; in particular $z_R=E_{(s,a)\sim
> \rho} [B(s,a)r(s,a)]$.

---

### Decision · Program_Chairs · 2021-09-27

**Decision:**

Accept (Poster)

**Comment:**

The majority of the reviews agrees that the paper presents an interesting new approach and votes for a clear accept. Accordingly, I recommend acceptance. As most reviewers also pointed out a few issues with the presentation and organisation of the paper, please take into account these comments when preparing the final version. Finally, one reviewer was not that completely convinced by the suggested representation. In the discussion (s)he said (I think expressing the point better than in the original review):

"My concern is that the representation carries too much information. To be specific, (please correct me if my statements are wrong), in their tabular MDP example, the function $F$ has a large continuous input space (dimension>SA) and thus requires function approximations and large computation power. In contrast, learning the MDP only involves $S^2A$ exact parameters, and the planning under a given MDP and reward function is usually not the bottleneck (using dynamic programming and given a discount factor in (0,1)). This issue can be even worse for general MDPs, and the paper seems to not discuss too much on the approximation power of such a representative for other MDPs."

Although I understand that the paper suggests not to learn a particular MDP but a range of tasks, the question whether the arising complexity of the final problem is reasonable when compared to the complexity of a set of single tasks is I think a valid one and should be discussed.